# Temporally-coordinated bivalent histone modifications of *BCG1* enable fungal invasion and immune evasion

Xiaozhen Zhao[1,5], Yiming Wang [1,5], Bingqin Yuan[1,5], Hanxi Zhao[1,5], Yujie Wang[1], Zheng Tan[1], Zhiyuan Wang[1], Huijun Wu[1], Gang Li [1], Wei Song[1], Ravi Gupta [2], Kenichi Tsuda [3], Zhonghua Ma [4], Xuewen Gao[1] & Qin Gu [1] ✉

Bivalent histone modifications, including functionally opposite H3K4me3 and H3K27me3 marks simultaneously on the same nucleosome, control various cellular processes by fine-tuning the gene expression in eukaryotes. However, the role of bivalent histone modifications in fungal virulence remains elusive. By mapping the genome-wide landscape of H3K4me3 and H3K27me3 dynamic modifications in *Fusarium graminearum* (Fg) during invasion, we identify the infection-related bivalent chromatin-marked genes (*BCGs*). *BCG1* gene, which encodes a secreted *Fusarium*-specific xylanase containing a G/Q-rich motif, displays the highest increase of bivalent modification during Fg infection. We report that the G/Q-rich motif of BCG1 is a stimulator of its xylanase activity and is essential for the full virulence of Fg. Intriguingly, this G/Q-rich motif is recognized by pattern-recognition receptors to trigger plant immunity. We discover that Fg employs H3K4me3 modification to induce *BCG1* expression required for host cell wall degradation. After breaching the cell wall barrier, this active chromatin state is reset to bivalency by co-modifying with H3K27me3, which enables epigenetic silencing of *BCG1* to escape from host immune surveillance. Collectively, our study highlights how fungal pathogens deploy bivalent epigenetic modification to achieve temporally-coordinated activation and suppression of a critical fungal gene, thereby facilitating successful infection and host immune evasion.

Epigenetic modifications, including histone modification, DNA methylation, and nucleosome positioning, are highly dynamic processes that control the intricate regulatory networks of gene expression[1–4]. Among these, histone lysine methylation has been implicated as a critical epigenetic switch that controls the expression of various genes associated with growth, development, and stress response in eukaryotes[5,6]. H3 lysine 4 trimethylation (H3K4me3) is generally a chromatin marker for transcriptional activation[7], while H3 lysine 27 trimethylation (H3K27me3) is associated with transcriptional silencing[8,9]. Although the regulatory mechanism is not clear,

[1]Department of Plant Pathology, College of Plant Protection, Nanjing Agricultural University, Key Laboratory of Monitoring and Management of Crop Diseases and Pest Insects, Ministry of Education, Nanjing, China. [2]College of General Education, Kookmin University, Seoul 02707, South Korea. [3]State Key Laboratory of Agricultural Microbiology, Hubei Hongshan Laboratory, Hubei Key Lab of Plant Pathology, College of Plant Science and Technology, Huazhong Agricultural University, Wuhan 430070, China. [4]State Key Laboratory of Rice Biology, the Key Laboratory of Biology of Crop Pathogens and Insects, Institute of Biotechnology, Zhejiang University, Hangzhou, China. [5]These authors contributed equally: Xiaozhen Zhao, Yiming Wang, Bingqin Yuan, Hanxi Zhao. ✉e-mail: guqin@njau.edu.cn

transcriptional reprograming directed by H3K4me3 or H3K27me3 modifications has been demonstrated to enable fungal pathogens to coordinate growth, metabolism, and development, thereby adapting to various host or natural environments[10–14].

"Bivalent chromatin domains" is a specific chromatin modification pattern that is simultaneously marked with both active (H3K4me3) and repressive (H3K27me3) histone modifications on the same nucleosomes[15,16]. H3K4me3-H3K27me3 bivalent domains were first identified in mouse embryonic stem cells (ESCs) and reported to poise the expression of lineage-specific genes associated with ESCs differentiation[15]. In plants, the H3K4me3-H3K27me3 bivalent modifications have only been identified in a few genomic regions in *Arabidopsis* seedings[17]. A recent study also revealed that *Arabidopsis* sperm cell has widespread apparent chromatin bivalency, which may play important roles in *Arabidopsis* male germline development[18]. In potatoes, bivalent H3K4me3-H3K27me3 modification was also observed and found to be potentially involved in the activation of stress response genes, but suppression of genes associated with developmental processes during cold stress[19]. Although these observations have supported the importance of bivalent H3K4me3-H3K27me3 modification in fine-tuning gene expression in both plants as well as animals, the presence and functions of the bivalent histone modifications in other organisms, especially in fungi, remain elusive.

Plant immunity is activated through the recognition of pathogen-associated molecular patterns (PAMPs) by pattern-recognition receptors (PRRs) in plants to initiate PAMP-triggered immunity (PTI), which is considered as the front line of defense. PTI is characterized by $Ca^{2+}$ influx, ROS-burst, MAPK-activation, and callose deposition that restrict the pathogen entry into the plant cells[20–22]. Adapted pathogens deploy various strategies to promote infection and evade host immunity. Emerging evidence suggests that pathogens can downregulate the production of these PAMPs by tightly regulating the expression of associated genes to escape the host immunity[23,24], however, the underlying mechanism involved in this fine-tuning of PAMP gene expression is poorly understood.

*Fusarium graminearum* (named Fg thereafter) is a devastating fungal pathogen that causes Fusarium head blight disease in the world[25,26]. In recent years, the wheat-Fg pathosystem has emerged as a model system for studying the mechanisms of host-pathogen interaction. We, therefore, utilized Fg-wheat pathosystem to dissect the role of bivalent histone modifications in plant-pathogen interaction. Our results reveal a remarkable decrease in both H3K4me3 and H3K27me3 modifications in Fg during infection; however, the co-modified regions were dramatically increased. Subsequently, we identified 11 infection-related bivalent chromatin-marked genes (BCGs) via sequential ChIP assay. Among them, *BCG1*, which encodes for a *Fuarium*-specific xylanase that enables successful infection of Fg through its xylanase activity, exhibited the highest bivalency during host invasion. Interestingly, the C-terminal G/Q-rich domain of BCG1, which is important for its xylanase activity, also mediates the interaction with pattern recognition receptor BAK1/SORBIR1 to trigger PTI in plants. We further show that a high H3K4me3 level on *BCG1* activates its expression to initiate host cell wall breakdown during early infection stages. Subsequently, upon breaching the cell wall barrier, the H3K27me3 was increased to establish H3K4me3-H3K27me3 bivalent modifications, thus repressing the expression of *BCG1* for host immune evasion. We also find that the absence of bivalent histone modification methyltransferases (Set1 and Kmt6) or its reader BP1 abolishes this precise regulation of *BCG1* expression, thus disrupting Fg pathogenesis. Taken together, these data contribute to the understanding of how fungal pathogen successfully invades host plants and evades host immunity through bivalent chromatin modification-based precisely epigenetic regulation of critical gene expression.

## Results

### Dynamic H3K4me3 and H3K27me3 histone modifications in Fg during infection

To gain insight into the genome-wide landscape of histone modifications dynamics during Fg infection, we mapped the H3K4me3 and H3K27me3 modifications of Fg during *in planta* (at 48 h post-inoculation (hpi) in wheat heads) and in vitro (0 hpi, cultured in Fusarium minimal medium) growth conditions (Supplementary Fig. 1a). Chromatin immunoprecipitation sequencing (ChIP-seq) assays revealed that H3K4me3 signals were distributed across all Fg chromosomes and were predominately enriched in gene bodies; meanwhile H3K27me3 signals were mainly found in the subtelomeric regions and were majorly enriched in the gene bodies and promoter regions in both in vitro and *in planta* cultured Fg (Supplementary Fig. 1b, c), which is consistent with previous reports[14,27]. Although H3K27me3 and H3K4me3 signals were mainly observed at distinct genomic regions under in vitro condition, significant overlapping signals of both H3K4me3 and H3K27me3 were detected *in planta* (Supplementary Fig. 1b and Fig. 1a; *p*-value = 3.1e$^{-15}$, binomial test). Further, histone modifications dynamics analysis showed that the global number of H3K4me3-only marked peaks/genes decreased significantly at 48 hpi during *in planta* growth condition as compared to in vitro, while the number of H3K27me3-only peaks/genes decreased to a lesser extent during infection (Fig. 1a). Interestingly, the number of H3K4me3-H3K27me3 co-marked peaks/genes was remarkably increased during *in planta* growth at 48 hpi (Fig. 1a), indicating the co-modifying of H3K4me3 and H3K27me3 is probably required for Fg infection.

Next, we performed RNA-seq analysis to investigate the transcriptional reprogramming of the epigenetically regulated genes in Fg during host invasion. Genes with H3K4me3-only signals were relatively highly expressed, whereas genes associated with H3K27me3-only regions showed relatively lower expression as compared to non-marked genes under both in vitro and *in planta* growth conditions (Supplementary Fig. 1d). These results were in line with the conventional roles of H3K4me3 and H3K27me3 as active and repressive chromatin marks in eukaryotes, respectively. Based on the profiles of these two histone modifications, 351 and 880 co-marked genes were identified in vitro and *in planta*, respectively (Fig. 1a), which displayed relatively higher transcriptional activity than the H3K27me3-only genes but lower than the H3K4me3-only marked genes (Supplementary Fig. 1d), suggesting a key role of H3K4me3-H3K27me3 co-modifying in transcriptional control of gene expression in Fg.

The coexistence of H3K4me3 and H3K27me3 modifications on the same nucleosomes is termed as bivalent histone marks[15,16]. Using DiffBind, 1105 and 153 genes (*p*-value < 0.005, log$_2$[fold change] ≥ 1.5) were identified with increased H3K4me3 and H3K27me3 modifications during infection, respectively (Fig. 1b). Among those, 42 genes were co-marked with both H3K4me3 and H3K27me3 modifications (Fig. 1b), indicating potential bivalent H3K4me3-H3K27me3 modifications. To examine whether the co-occurring of H3K4me3 and H3K27me3 marks were truly deposited on the same nucleosomes, sequential ChIP-seq on mononucleosomes was performed by first ChIP with anti-H3K4me3 antibody and followed by second ChIP with anti-H3K27me3 antibody using the same samples for ChIP-seq analyses. The results showed that among 42 co-marked genes, 16 genes were bivalently modified during infection (Fig. 1b). Other 26 co-marked genes may have H3K4me3 and H3K27me3 marks in different cells and may not be co-modified simultaneously on the same nucleosome.

### Bivalent chromatin-marked gene 1 (*BCG1*) plays critical roles in Fg pathogenesis

To uncover the regulatory roles of bivalent histone modifications in fungal pathogenesis, the temporal expression patterns of these 16

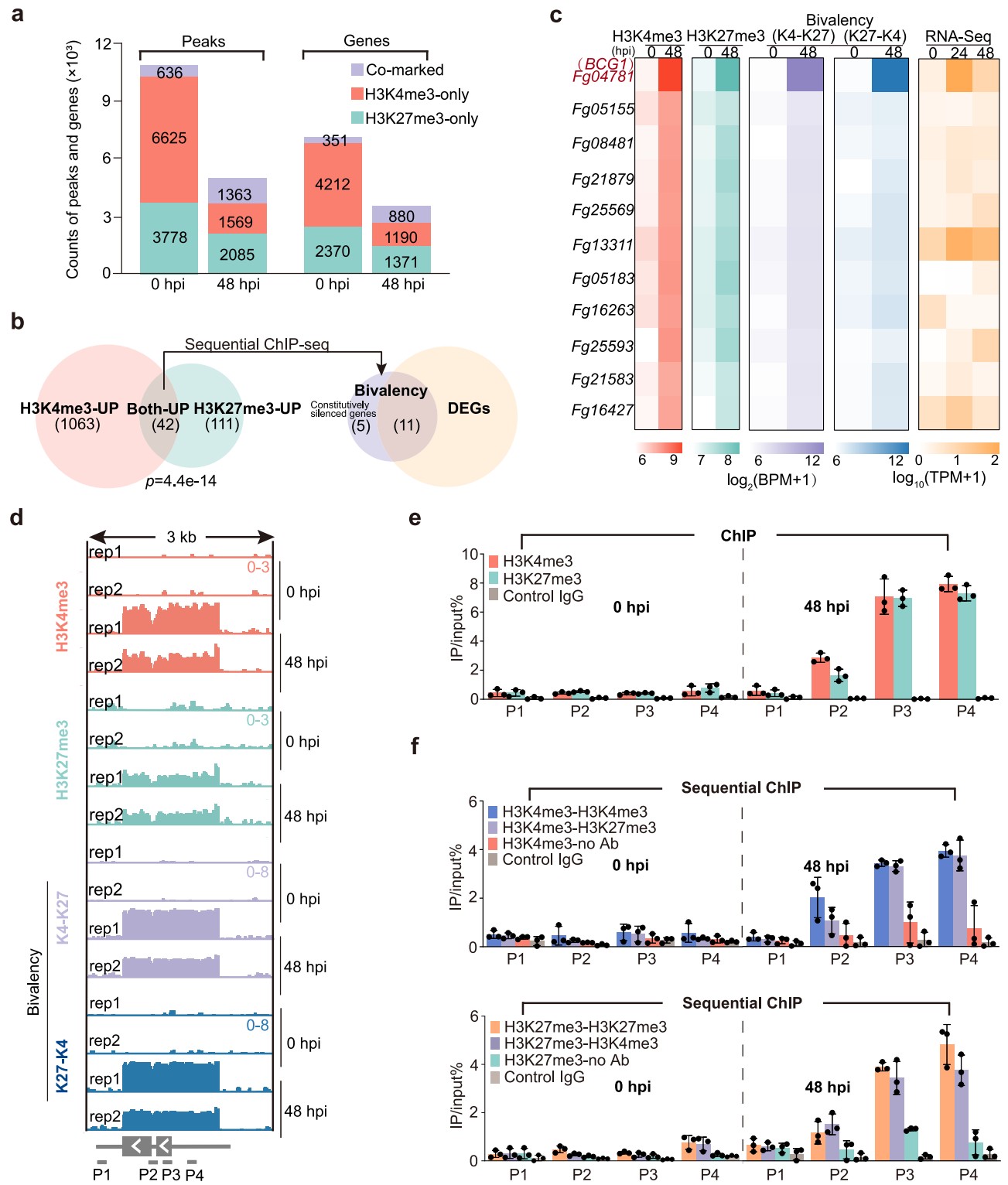

bivalent chromatin-marked genes (*BCGs*) were examined during different stages of Fg infection (at 0, 24, and 48 hpi). RNA-seq analyses revealed that 5 of 16 *BCGs* were constitutively silenced, while 11 genes exhibited transcriptional reprogramming at different infection stages (Fig. 1c). We then generated the independent deletion mutants of these 11 infection-induced *BCGs* to establish their roles in Fg infection (Supplementary Fig. 2, 3). None of these mutants exhibited phenotypes of altered fungal growth or stress tolerance (Supplementary Fig. 4); however, five mutants (*ΔFg04781*, *ΔFg08481*, *ΔFg05155*, *ΔFg21583*, and *ΔFg16247*) exhibited significantly reduced virulence on

wheat heads (Supplementary Fig. 5), highlighting a key role of H3K4me3-H3K27me3 bivalent modification in the regulation of Fg pathogenesis.

Among the 11 infection-induced *BCGs*, *FgO4781* (named *BCG1* thereafter) showed the strongest impact on Fg pathogenesis (Supplementary Fig. 5), and exhibited the highest increase in bivalent signals during host invasion (Fig. 1c, d). Using ChIP-qPCR and sequential ChIP-qPCR, we further determined the histone methylation dynamics at the *BCG1* locus during Fg infection. Four pairs of primers located in the vicinity of the *BCG1* locus were used (Fig. 1d). Consistent with the

**Fig. 1 | *BCG1* is predominantly marked by bivalent histone modifications during *F. graminearum* infection. a** The global number of H3K4me3-only, H3K27me3-only, and co-marked peaks and genes in *F. graminearum* at 0 and 48 hpi. **b** Venn diagram showing the overlap between the genes with increased H3K4me3 and H3K27me3 histone modifications during *F. graminearum* infection (left panel). Overlap of genes with increased bivalent modification and differential expression during infection (right panel). Bioinformatics analysis filtered the genes with increased bivalent modification and differential expression during *F. graminearum* infection. **c** Heatmaps showing the enrichment density of H3K4me3, H3K27me3, and bivalent modifications of these 11 bivalent chromatin modified genes (0 and 48 hpi) (left), as well as their corresponding transcription patterns (0, 24, and 48 hpi) (right). The functional gene *BCG1* (*FgO4781*) is labeled in red. **d** Browser view of H3K4me3, H3K27me3, and bivalency densities around *BCG1* locus (3 kb around *BCG1*) at 0 and 48 hpi. ChIP-seq and sequential ChIP-seq signals were normalized by

input using BPM. Gray boxes represent the open reading frames of *BCG1* gene. Four pairs of primers (P1, P2, P3, and P4) were used to quantify H3K4me3, H3K27me3, and bivalency deposition across the *BCG1* gene. K4-K27 in c and d represents sequential ChIP conducted by first ChIP using an anti-H3K4me3 antibody followed by second ChIP using an anti-H3K27me3 antibody, while K27-K4 represents sequential ChIP performed using antibodies in the reverse order. **e** ChIP-qPCR measurements of H3K27me3 and H3K4me3 enrichments around *BCG1* locus. **f** Sequential ChIP-qPCR showing the H3K4me3-H3K27me3 bivalent modification across *BCG1* gene. Altering the order of indicated antibodies in two rounds of ChIP shows similar results in sequential ChIP-qPCR. No Ab represents no antibody in the second ChIP. Control IgG represents the control for the ChIP specificity. Relative accumulation levels were represented by percentage of input. Values are the means ± standard deviation (*n* = 3, biologically independent experiments). Source data are provided as a Source Data file.

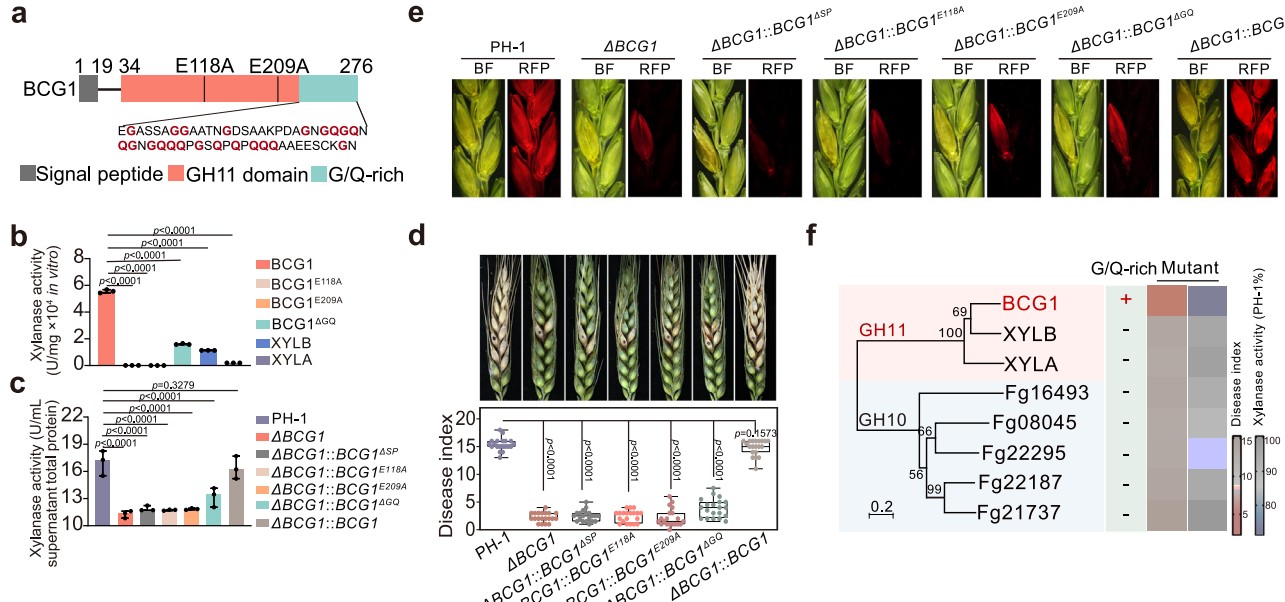

**Fig. 2 | BCG1 is required for *F. graminearum* virulence. a** Domain architecture of *F. graminearum* BCG1. **b** In vitro assays for the xylanase activity of indicated recombinant proteins. **c** The extracellular xylanase activities are determined on the total proteins from the supernatants of PH-1, *ΔBCG1*, and the complementary strains (*ΔBCG1::BCG1*, *ΔBCG1::BCG1^ΔSP^*, *ΔBCG1::BCG1^ΔGQ^*, *ΔBCG1::BCG1^E118A^*, and *ΔBCG1::BCG1^E209A^*) cultured in 1.1% CMC medium. In (**b**) and (**c**), the values are the means ± standard deviation (*n* = 3, biologically independent experiments). Statistical significance was assessed using a one-way ANOVA with Fisher's LSD test (*p* value < 0.01). **d** Virulence of PH-1 (WT), *ΔBCG1*, and the complementary strains were evaluated on wheat heads. Representative images of infected wheat heads were photographed at 14 days post-inoculation (dpi). Quantification was estimated with data obtained from 20 biologically independent samples. For the box plot, the

center line indicates the median; the upper and lower bounds indicate the 75th and 25th percentiles, respectively; and the whiskers indicate the minimum and maximum. Statistical significance was assessed using a one-way ANOVA with Fisher's LSD test (*p*-value < 0.01). **e** Cross-sections of inoculated and adjacent wheat spikelets that were inoculated with PH-1, the mutants, and the complementary strains bearing FgActin-RFP. The samples were taken at 7 dpi. BF, bright field; RFP, red fluorescent protein. **f** Phylogeny of GH10 and GH11 members from *F. graminearum*. The plus (+) and minus (−) symbols indicate the presence and absence of the G/Q-rich motif, respectively. Bootstrap percentage for each branch is indicated. The scale bar represents 20% weighted sequence divergence. Blue and red colors represent relatively low xylanase and virulence activities, respectively. Source data are provided as a Source Data file.

ChIP- and sequential ChIP-seq data, we found a high accumulation of H3K4me3, H3K27me3, as well as bivalent signals across the promoter and gene body regions of *BCG1* specifically under *in planta* condition, while weaker signals were detected in vitro (Fig. 1e, f). Moreover, RNA-seq data revealed that *BCG1* was strongly induced at 24 hpi, but markedly declined at 48 hpi (Fig. 1c), implying that bivalent histone modification-based modulation of *BCG1* expression may play a pivotal role in Fg infection.

### *BCG1* encodes a xylanase that enables successful infection of Fg through its host cell wall degrading activity

Microbial xylanases (endo-1,4-β-xylanase; EC 3.2.1.8) are the members of glycoside hydrolase (GHs) families, mainly GH10 and GH11, and are known to be key components of pathogen virulence[28,29]. Sequence

analysis of BCG1 revealed that it contains a typical N-terminal signal peptide and a GH11 domain (Fig. 2a), suggesting that BCG1 is a secreted xylanase. To characterize the xylanase activity of BCG1, we expressed BCG1 protein, as well as its catalytic-site mutated versions BCG1^E118A^ and BCG1^E209A^ in *Pichia pastoris* (Supplementary Fig. 6a). In vitro hydrolase activity of recombinant BCG1 protein showed a high xylanase activity, which was completely abolished in BCG1^E118A^ and BCG1^E209A^ proteins (Fig. 2b). Consistently, we also found that BCG1, but not BCG1^E118A^ and BCG1^E209A^ could degrade the xylan derived from wheat palea (Supplementary Fig. 7a). Moreover, *ΔBCG1* mutant displayed a significant decreased extracellular xylan-degrading activity as compared to the wild-type PH-1 (Fig. 2c), indicating that BCG1 contributes greatly to the secreted xylanase activity of Fg. Transmission electron microscopy (TEM) observation also showed that *ΔBCG1* was deficient in breaking

through the cell walls of wheat palea at 2 days post-inoculation (dpi) (Supplementary Fig. 7b). In line, *ΔBCG1* exhibited significantly reduced virulence on wheat heads (Fig. 2d, e), suggesting that BCG1 xylanase activity is required for Fg virulence. To further establish the role of BCG1 in Fg virulence, we complemented the *ΔBCG1* strain with a full-length *BCG1*, which reversed the reduced xylanase activity of *ΔBCG1* and exhibited similar virulence as wild-type PH-1. However, *ΔBCG1* complemented either with the signal peptide mutated *BCG1^ΔSP^*, or catalytic-site mutated *BCG1^E118A^* and *BCG1^E209A^* driven by *BCG1* native promoter, still exhibited similar defects in extracellular xylanase activity and fungal virulence (Fig. 2c−e and Supplementary Fig. 7c). Taken together, our results demonstrate that the secreted xylanase activity of BCG1 is required for the full virulence of Fg. Notably, *ΔBCG1::BCG1^ΔSP^*, *ΔBCG1::BCG1^E118A^*, and *ΔBCG1::BCG1^E209A^* strains showed normal filamentous growth (Supplementary Fig. 7d), suggesting that BCG1-mediated xylanase activity is only required for the Fg virulence and it does not affect in vitro growth of the fungus.

Genome-wide prediction showed the presence of five and three members of GH10 and GH11 family proteins in Fg including BCG1 (Fig. 2f). To understand whether those GH family proteins are also required for Fg virulence, we generated the deletion mutants of those genes. Our results revealed that *ΔBCG1* showed markedly stronger defects in both extracellular xylanase activity and fungal virulence than other xylanase mutants (Fig. 2f and Supplementary Fig. 8a−d), indicating that BCG1 is the major xylanase required for Fg virulence. In addition to the BCG1 protein, its two paralogs of the GH11 family proteins, XYLA and XYLB, were also expressed and purified from *P. pastori* (Supplementary Fig. 6a). Although XYLA and XYLB contain the conserved residues critical for enzyme activity and exhibit high amino acid sequence similarities (59% and 52%, respectively) with BCG1, they displayed dramatically decreased xylanase activity than that of BCG1 (Fig. 2b). Sequence alignment showed that BCG1 contains a previously unrecognized C-terminal G/Q-rich motif (Fig. 2a and Supplementary Fig. 9a), which is absent in BCG1 paralogs and members of GH10 xylanase proteins (Fig. 2f), implying that this G/Q-rich motif might contribute to higher xylanase activity of BCG1. To verify this hypothesis, we expressed and purified the truncated protein BCG1^ΔGQ^, lacking the G/Q-rich domain, from *P. pastoris* (Supplementary Fig. 6a). In vitro hydrolase activity assays showed that BCG1^ΔGQ^ protein displayed a significantly reduced xylanase activity (Fig. 2b). Moreover, complementation of *ΔBCG1* with *BCG1^ΔGQ^* expressing BCG1^ΔGQ^ under the native promoter (*ΔBCG1::BCG1^ΔGQ^*), failed to recover the reduced extracellular xylanase activity and fungal virulence as observed in the case of *ΔBCG1::BCG1* complementary strain (Fig. 2c−e and Supplementary Fig. 7c), indicating that G/Q-rich motif of BCG1 is a stimulator of its xylanase activity and is required for the full virulence of Fg.

## BCG1 functions as a PAMP in plants

Plant innate immune system can sense the presence of PAMPs to initiate PTI responses that restrict pathogen infection[30–32]. We noticed that infiltration of 0.1 to 10 μM recombinant BCG1 proteins into the mesophyll of *N. benthamiana* leaves showed BCG1-mediated cell death in a dosage-dependent manner (Supplementary Fig. 10a). Moreover, transiently expressed HA-tagged BCG1 also exhibited this cell death activity in *N. benthamiana*, however, its signal peptide deletion deleted version BCG1^ΔSP^, failed to exhibit such effect (Fig. 3a, b and Supplementary Fig. 10b, c). The cell death lacking effect of BCG1^ΔSP^ was reversed when it was expressed with a pathogenesis-related protein 1 (PR1) signal peptide at its N-terminus, PR1-SP-BCG1^ΔSP^ (Supplementary Fig. 10b, c). These results showed that the apoplastic localization of BCG1 is a prerequisite for its cell death activity, suggesting a potential role of BCG1 as a PAMP. Transient expression of BCG1 in *N. benthaminana* resulted in an accumulation of reactive oxygen species (ROS) and deposition of callose, the hallmarks of PTI, and greatly enhanced resistance to *Phytophthora capsici* (Supplementary Fig. 10d−f).

Notably, we further observed that more than 250 nM concentration of BCG1 protein was required to trigger a significant ROS burst in *N. benthamiana* leaf disks (Fig. 3c), indicating it to be a threshold concentration of BCG1 for immune activation.

We next examined whether BCG1-triggered plant immunity via BAK1/SERK3 and SOBIR1, the two central membrane localized PRRs[33]. We generated *BAK1*- and *SOBIR1*-silenced *N. benthamiana* plants by tobacco rattle virus (TRV)-based VIGS, which exhibited approximately 70% reduction in *BAK1* and *SOBIR1* expression as compared to that in TRV:*GFP* plants (Supplementary Fig. 11a). Notably, the BCG1-induced cell death was totally abolished in both *BAK1*- and *SOBIR1*-silenced plants, but not TRV:*GFP* control plants, which is consistent with the positive control INF1, key oomycete PAMP that triggers cell death through recognition by BAK1 and SOBIR1 (Supplementary Fig. 11b, c). Furthermore, ROS burst induced by recombinant BCG1 protein was also dramatically compromised in *BAK1*- or *SOBIR1*-silenced *N. benthamiana* leaves (Fig. 3d). In addition to the *N. benthamiana*, BCG1 was also able to induce ROS burst and expression of defense marker genes, such as *FRK1* and *PR1*, in *Arabidopsis thaliana* wild-type Col-0, but not in *bak1/bkk1/cerk1* triple mutant (*bbc*), which is deficient in PAMP recognition[34] (Fig. 3e, f). Taken together, these data illustrate that BCG1 functions as a PAMP of Fg and is recognized by plant PRRs BAK1/SERK3 and SOBIR1.

In line with the observation that BCG1^E118A^, xylanase-dead version protein, could trigger cell death similar to the BCG1 (Fig. 3a), BCG1^E118A^ also induced a comparable level of ROS with BCG1 in *N. benthamiana* and *A. thaliana* (Fig. 3d, f). In addition, similar to BCG1, BCG1^E118A^-triggered ROS burst and the expression of *FRK1* and *PR1* were abolished in both *BAK1* and *SOBIR1*-silenced *N. benthamiana* leaves (Fig. 3d), and *A. thaliana bbc* mutant (Fig. 3e, f). Collectively, these results indicate that BCG1-triggered plant immune responses are independent of its xylanase activity.

## G/Q-rich motif is specifically conserved in the *Fusarium* genus and is required for the biological functions of BCG1

To explore the possible contribution of G/Q-rich motif to the cell death inducing function of BCG1, we transiently expressed BCG1^ΔGQ^ and BCG1^ΔGQ/E118A^ (loss of both G/Q-rich motif and xylanase activity) in *N. benthamiana*. Both BCG1^ΔGQ^ and BCG1^ΔGQ/E118A^ exhibited significantly reduced cell-death activity as compared to BCG1 (Fig. 3a), indicating that the G/Q-rich motif is required for the full cell death inducing ability of BCG1. In addition, both BCG1^ΔGQ^ and BCG1^ΔGQ/E118A^ were unable to induce ROS burst in *N. benthamiana* and *A. thaliana* (Fig. 3d, f) and failed to induce *PR1* and *FRK1* expression in *A. thaliana* (Fig. 3e), suggesting that G/Q-rich motif is also required for BCG1-triggered plant immunity. Since BAK1 and SOBIR1 were required for BCG1-triggered plant immunity, we further evaluated the associations between BCG1 and these two PRRs. Co-immunoprecipitation (Co-IP) assays showed that BCG1 and BCG1^E118A^ could interact with both BAK1 and SOBIR1; however, these interactions were dismissed in the absence of G/Q-rich motif (Fig. 3g, h). These findings collectively demonstrate that G/Q-rich motif is required for the BCG1-triggered immune response.

Next, we investigated the phylogenetic distribution of G/Q-rich motif in the organisms with published genomes. Interestingly, this G/Q-rich motif was found to be exclusively present in the GH11 family proteins from *Fusarium* genus only (Fig. 4a). We, therefore, cloned BCG1 orthologs from other *Fusarium* species, including *Fusarium oxysporum* (FoBCG1) and *Fusarium verticillioides* (FvBCG1) and generated their recombinant proteins in *P. pastoris* (Supplementary Fig. 6b). Both FoBCG1 and FvBCG1 proteins showed strong xylanase activity and ROS burst induction ability (Fig. 4b), indicating that BCG1 orthologs may be functionally conserved across various *Fusarium* species. To further determine whether BCG1 activity in other *Fusarium* species requires G/Q-rich motif, we generated the truncated proteins FoBCG1^ΔGQ^ and FvBCG1^ΔGQ^, which lack the G/Q-rich motif

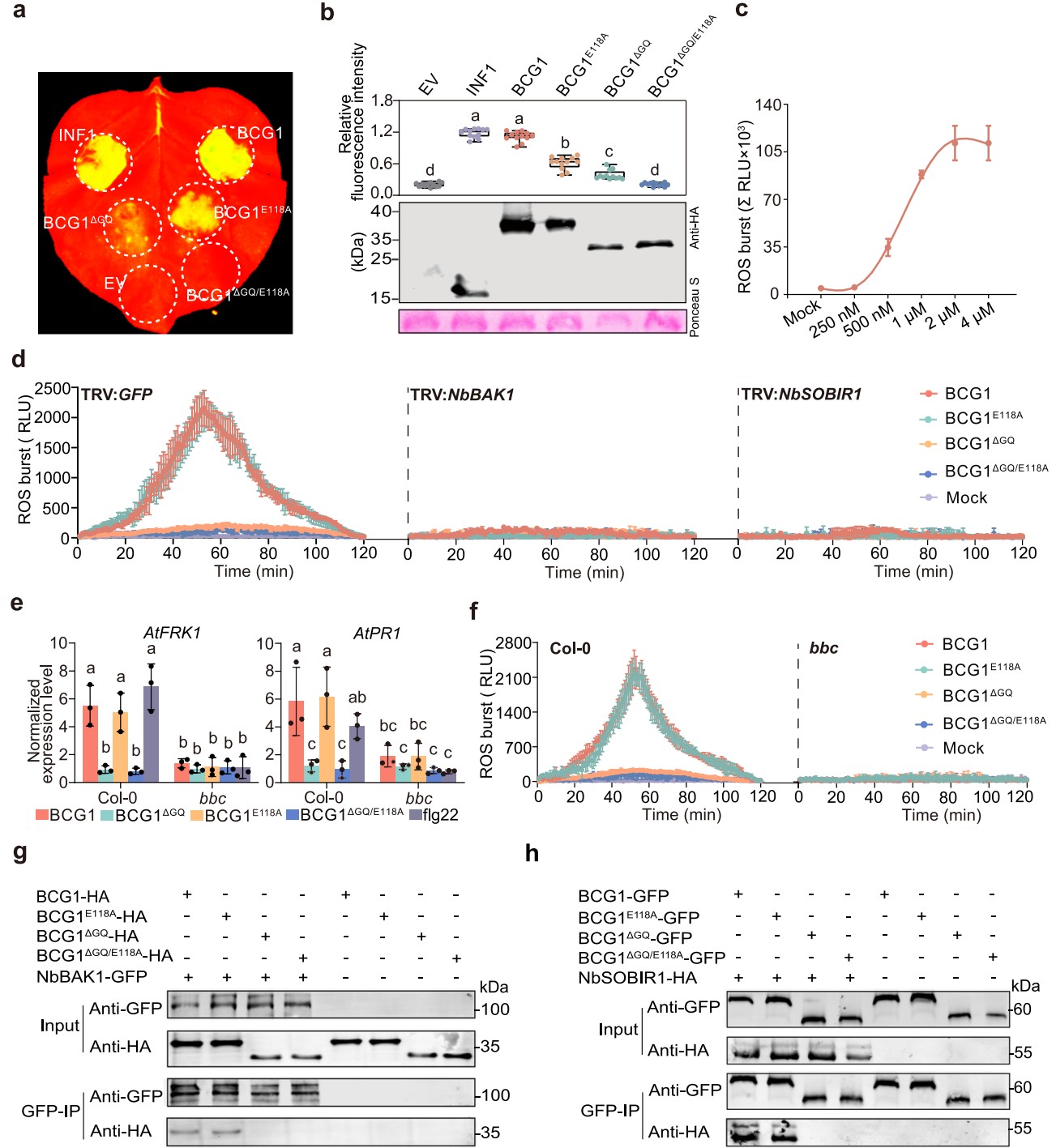

**Fig. 3 | BCG1-triggered plant immunity requires G/Q-rich domain.**
**a** Representative *N. benthamiana* leaves showing cell death at 5 days post-inoculation (dpi) with *Agrobacterium* strains expressing indicated proteins under UV light. EV, empty vector. **b** Cell death intensity is indicated by the relative quantitative fluorescence intensity that is normalized to the INF1. Quantification was estimated with data obtained from twelve biologically independent samples. For the box plot, the center line indicates the median; the upper and lower bounds indicate the 75th and 25th percentiles, respectively; and the whiskers indicate the minimum and maximum. Different letters denote significant differences (*p* value < 0.01, one-way ANOVA) (upper panel). Western blot analysis showing the expressed proteins using the anti-HA antibody. Ponceau S staining serves as loading control (lower panel). **c** Dose−response curves of total ROS production induced by BCG1 protein in *N. benthamiana* leaf discs over 120 min. The values are the means ± standard deviation (SD) (*n* = 3, biologically independent experiments).

**d** ROS production induced by 1 μM of indicated recombinant proteins in silenced *N. benthamiana* leaves. The values are the means ± SD (*n* = 3, biologically independent samples). **e** Relative expression of defense-related genes *AtFRK1* and *AtPR1* in leaves of Col-0 and *bak1 bkk1 cerk1* (*bbc*) mutant plants of *A. thaliana* was evaluated by RT-qPCR using *ACTIN2* gene as the internal control. Values represent the means ± SD (*n* = 3, biologically independent experiments). Different letters denote significant differences (*p* value < 0.01, one-way ANOVA). *P* values for (**b, e**) are shown in the Source Data. **f** ROS burst induced by 1 μM of indicated recombinant proteins in Col-0 and *bbc* mutant leaves of *A. thaliana*. The values are the means ± SD (*n* = 3, biologically independent samples). **g**−**h** GFP- or HA-tagged BCG1, BCG1^E118A, BCG1^ΔGQ, and BCG1^ΔGQ/E118A was co-expressed with HA- or GFP-tagged NbBAK1 and NbSORIR1 in *N. benthamiana* leaves. Co-IP assays were performed using GFP-trap A beads and the indicated proteins were immunoblotted with anti-HA and anti-GFP antibodies, respectively. Source data are provided as a Source Data file.

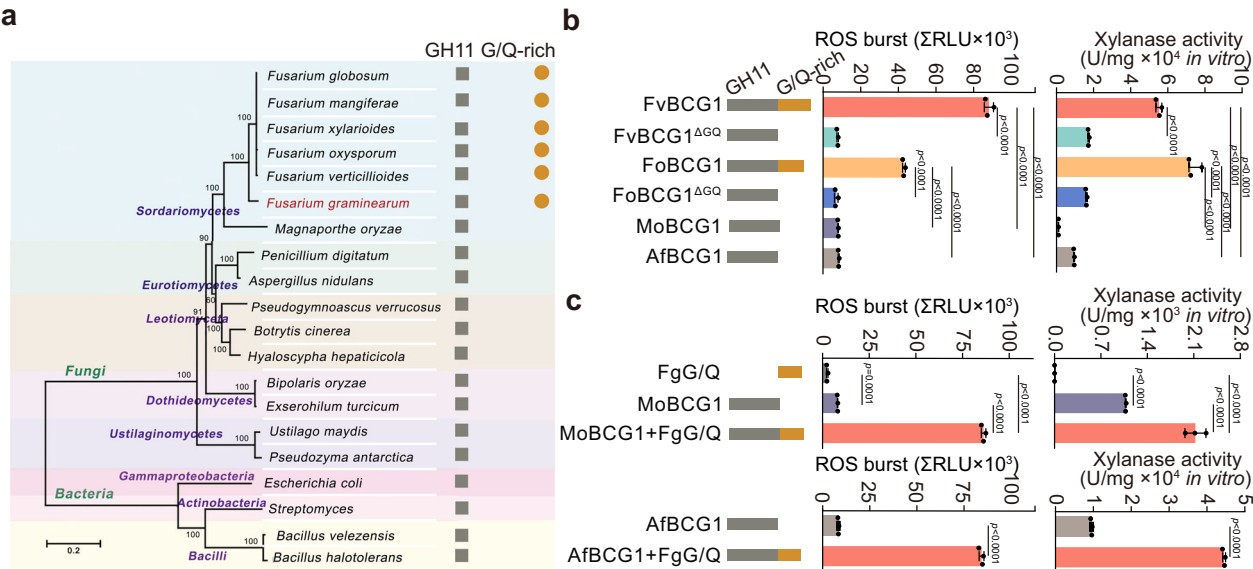

**Fig. 4 | G/Q-rich motif is specifically conserved in the *Fusarium* genus and primes the PAMP and xylanase activity of BCG1. a** G/Q-rich motif is specifically present in *Fusarium* genus. The phylogenetic tree was constructed based on the amino acid sequences of RPB2 from 16 fungal species and 4 bacterial species with Mega 7.0 using the neighbor-joining method (left). The presence of GH11 domain and G/Q-rich motif is indicated with brown squares and circles, respectively (right). **b** ROS burst induced by indicated proteins in *N. benthamiana* leaves. Total relative luminescence units (RLU) were detected over 120 min (left). Xylanase activity of the

indicated proteins was measured following a 2,4-dinitrosalicylic acid (DNS) assay (right). **c** G/Q-rich motif of *F. graminearum* (FgG/Q) confers increased ROS-inducing and xylanase activity of BCG1 orthologs in *M. oryzae* and *A. flavus* (MoBCG1 and AfBCG1). Data in **b** and **c** presented are the means ± standard deviation ($n = 3$, biologically independent experiments). Statistical significance was assessed using a one-way ANOVA with Fisher's LSD test ($p$-value < 0.01). Source data are provided as a Source Data file.

(Supplementary Fig. 6b). Our results showed that loss of G/Q-rich motif led to the compromised ROS production and significantly reduced xylanase activity (Fig. 4b), which was consistent with the findings observed in Fg. Moreover, BCG1 orthologs derived from *Magnaporthe oryzae* (MoBCG1) and *Aspergillus flavus* (AfBCG1) without G/Q-rich motif also exhibited markedly reduced ROS production and xylanase activity (Fig. 4b), further indicating a critical role of G/Q-rich motif in ROS production and xylanase activity of BCG1. We next investigated whether the individual G/Q-rich motif is able to exhibit these activities without the BCG1 protein. We cloned and purified G/Q-rich motif from *P. pastoris* (Supplementary Fig. 6c). Interestingly, although G/Q-rich motif exhibited no xylan-degrading activity and failed to trigger the BAK1/SOBIR1-dependent plant immune responses (Fig. 4c and Supplementary Fig. 9b), when we fused the Fg-derived G/Q-rich motif with MoBCG1 and AfBCG1 (MoBCG1+FgG/Q and AfBCG1+FgG/Q), a significantly increased xylanase activity and ROS burst induction ability of the fused proteins were observed (Fig. 4c). Taken together, these findings suggest the importance of G/Q-rich motif in the biological function of GH11 family proteins and demonstrate that the *Fusarium*-specific G/Q-rich motif plays critical roles in the PAMP and xylanase activity of BCG1.

## BCG1 triggers plant immunity in wheat

To determine whether BCG1 triggers plant immunity in the Fg host plant wheat, we overexpressed BCG1 in PH-1 under the control of *gpda*, a strong constitutive promoter from *Aspergillus nidulans*[35]. The expression of BCG1 was increased by over 20-fold in the BCG1 overexpression strain (*OE-BCG1*) as compared to the PH-1 (Supplementary Fig. 12a). Notably, *OE-BCG1* showed normal growth morphology (Supplementary Fig. 12b); however, its virulence was markedly reduced on wheat heads and seeding leaves (Fig. 5a, b and Supplementary Fig. 12c). Wheat seedling leaves inoculated with *OE-BCG1* showed significantly increased ROS accumulation as compared to the PH-1 (Fig. 5c, d). Moreover, we observed that more than 250 nM concentration of BCG1 pretreatment was required to significantly reduce

the disease development by PH-1 strain (Supplementary Fig. 12d), indicating that there is also a threshold of BCG1 concentration to induce wheat defense against Fg invasion. RNA-seq analysis further showed that the BCG1-induced wheat genes were related to defense response and immune system process (Fig. 5f, g), confirming the role of BCG1 in the induction of immune responses in wheat.

The BCG1^E118A, BCG1^ΔGQ, or BCG1^ΔGQ/E118A-overexpressing Fg strains (namely *OE-BCG1^E118A*, *OE-BCG1^ΔGQ*, and *OE-BCG1^ΔGQ/E118A*, respectively) were further generated. Compared to PH-1, *OE-BCG1^E118A* strain induced higher $H_2O_2$ accumulation (Fig. 5c, d) and significantly reduced virulence (Fig. 5a, b and Supplementary Fig. 12c) in wheat. By contrast, both *OE-BCG1^ΔGQ* and *OE-BCG1^ΔGQ/E118A* strains showed no obvious changes in virulence and induced comparable levels of ROS with PH-1 (Fig. 5a–d and Supplementary Fig. 12c). In line, pretreatment with BCG1^E118A protein in wheat seedling leaves led to the reduced disease symptoms and enhanced expression of defense-related genes (*TaPR1*, *TaERF113*, and *TaWRKY28*) expression. In contrast, BCG1^ΔGQ and BCG1^ΔGQ/E118A proteins did not affect wheat defense against Fg infection and expression of these defense genes (Fig. 5e, h), confirming that BCG1-triggered immune responses in wheat are dependent on its G/Q-rich motif and are independent of its xylanase activity. In 2 years filed trials (2021 and 2022), pretreatment of both BCG1 and BCG1^E118A proteins by foliar spray on wheat heads showed control efficacy of 40–70% against Fusarium head blight (FHB) (Fig. 5i), indicating the application potential of BCG1 proteins as elicitors for FHB management.

## Temporally-coordinated bivalent histone modifications of *BCG1* enable fungal invasion and immune evasion

Previous studies have shown that Fg breaches the host cell wall barrier to initiate its infection and subsequently uses a biotrophic strategy to successfully invade the host cell[36,37] (Fig. 6a). Here, we observed the infection process of Fg on wheat seedling leaves by using scanning electron microscopy (SEM), TEM, and live-cell confocal microscopy. We showed that Fg established surface colonization and began to

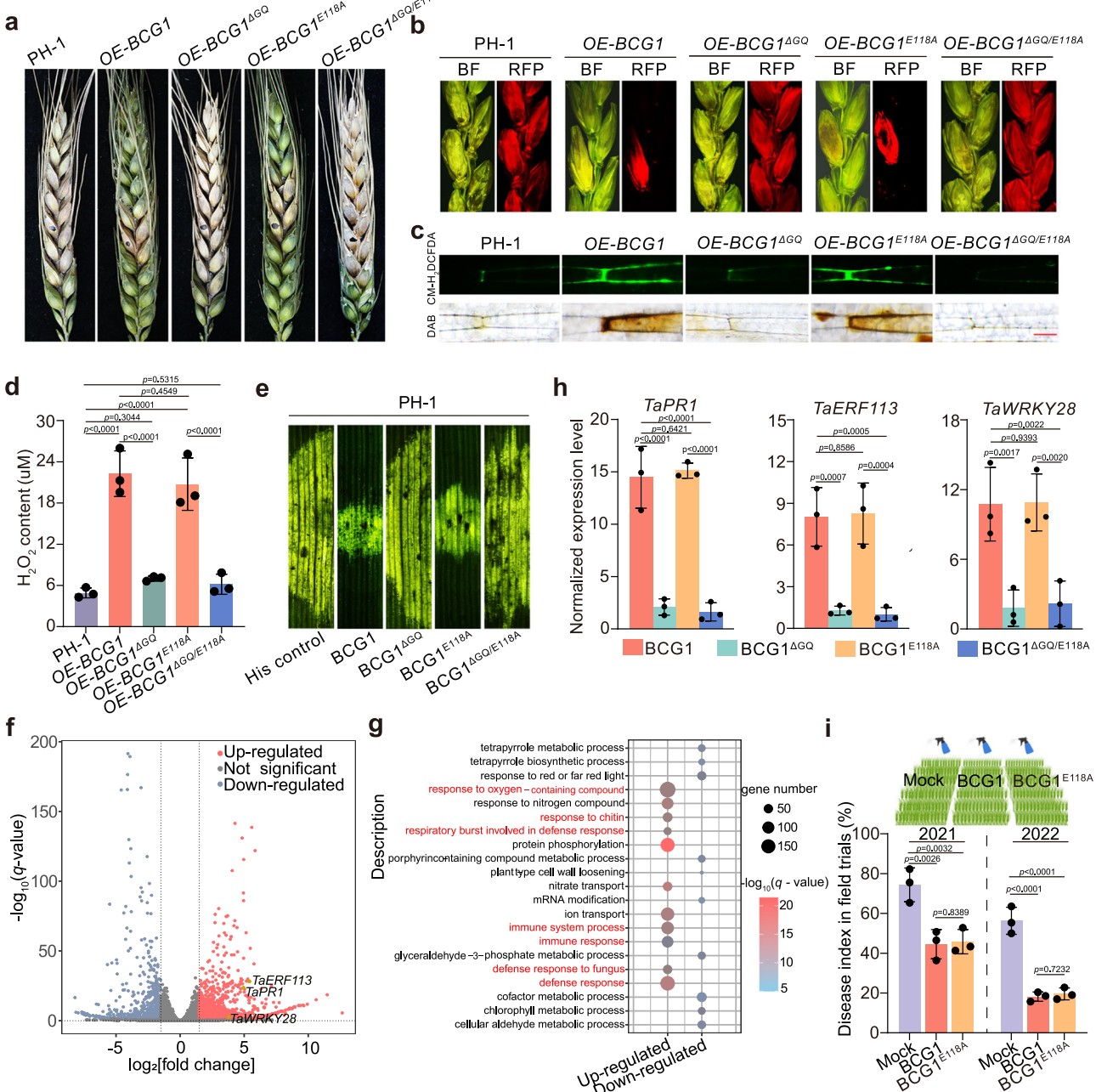

**Fig. 5 | BCG1 triggers plant immunity in wheat. a** Virulence of PH-1 and the indicated overexpression strains were evaluated on wheat heads. Representative images of infected wheat heads were photographed at 14 days post-inoculation (dpi). **b** Cross-sections of inoculated and adjacent wheat spikelets inoculated with PH-1 and the overexpression strains expressing FgActin-RFP. The samples were taken at 7 dpi. BF, bright field; RFP, red fluorescent protein. **c** DCFH-DA- (green fluorescent) and DAB- (dark brown color) staining showing ROS ($H_2O_2$) accumulation in wheat seedling leaves 48 h post-inoculation (hpi) with PH-1 and the indicated overexpression strains. Bar = 50 μm. **d** Quantification of $H_2O_2$ production in wheat seedling leaves at 48 h post-inoculation (hpi). **e** Disease resistance of wheat seedling leaves to *F. graminearum* wild-type PH-1 triggered by indicated proteins. Wheat seedling leaves were pretreated with 1 μM of indicated proteins 24 h before *F. graminearum* inoculation. Representative images of infected wheat seedling leaves were photographed at 3 dpi. **f** Comparative transcriptome analysis showing the BCG1-induced differentially expressed genes (DEGs) in wheat seeding leaves.

DEGs were defined as $log_2$(BCG1/mock fold change) ≥1.5 or ≤−1.5 and *q* value < 0.05. The defense-related genes, *TaPR1*, *TaERF113*, and *TaWRKY28*, are highlighted in yellow. **g** The top 21 gene ontology (GO) categories within biological processes enriched in BCG1-induced genes. Note that genes involved in defense responses and immune system processes were up-regulated upon BCG1 treatment. **h** Relative expression of defense-related genes in wheat leaves after treatment with 1 μM of indicated recombinant proteins was evaluated by RT-qPCR. *TaGAPDH* serves as an internal control. **i** Schematic representation of exogenous spray of indicated proteins on wheat heads in the field (upper panel). The disease index of Fusarium head blight after pretreatment with 1 μM of BCG1 and BCG1^E118A proteins in field trials conducted in 2021 and 2022. 1 μM of His protein was used for the untreated control (Mock). In (**d, h, i**) data presented are the means ± standard deviation (*n* = 3, biologically independent samples). Statistical significance was assessed using a one-way ANOVA with Fisher's LSD test (*p* value < 0.01). Source data are provided as a Source Data file.

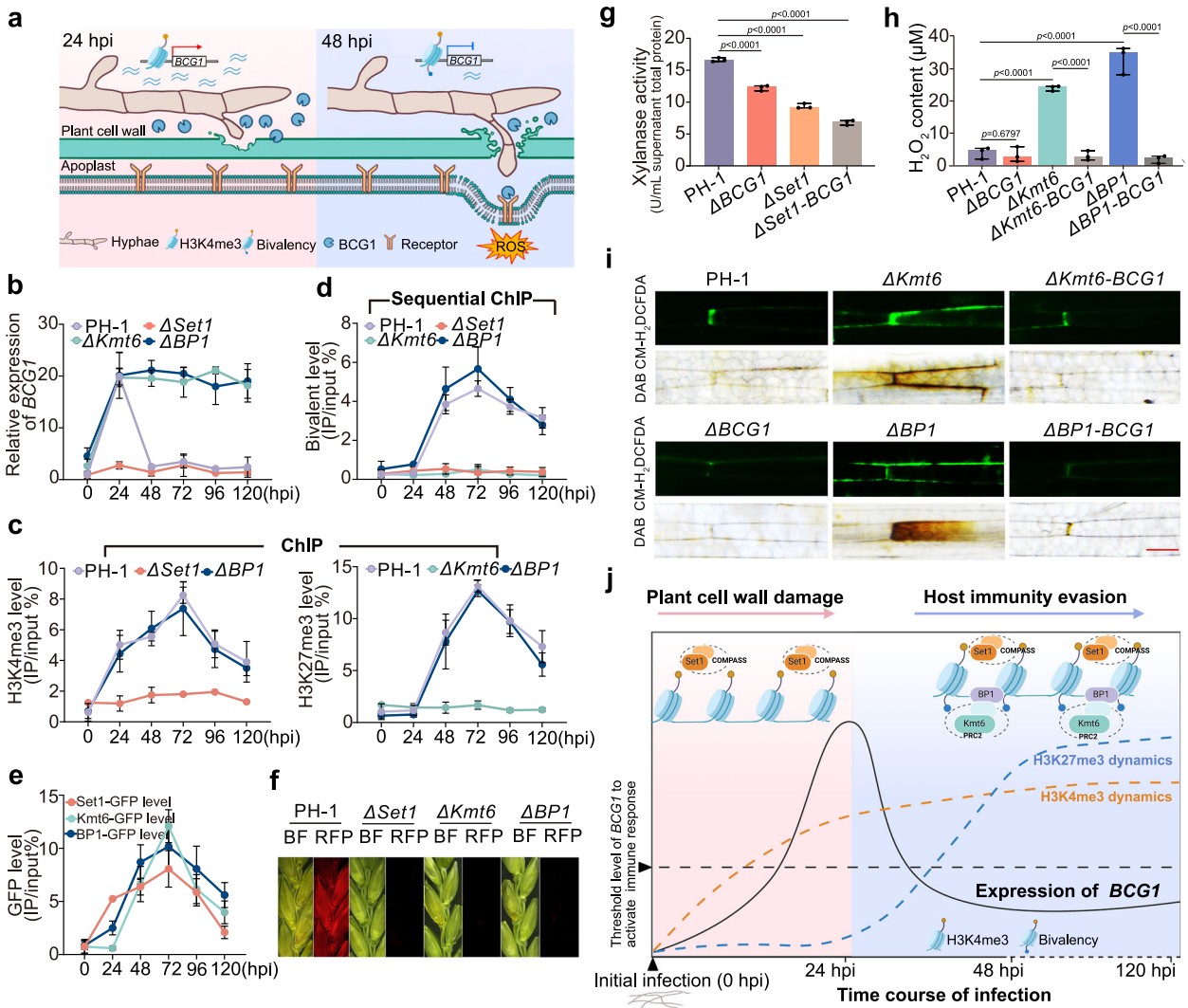

**Fig. 6 | Temporally-coordinated bivalent histone modifications of *BCG1* enable fungal invasion and immune evasion. a** Schematic diagram of the bivalent epigenetic regulation of *BCG1* during wheat infection. Figure created using Procreate (https://procreate.com/). **b** Relative expression level of *BCG1* in wild-type PH-1 and the indicated mutants at 0–120 h post-inoculation (hpi) were evaluated by RT-qPCR. **c** ChIP-qPCR measurements of H3K4me3 and H3K27me3 deposition at *BCG1* in wild-type PH-1 and the mutants (*ΔSet1*, *ΔKmt6*, and *ΔBP1*) at 0–120 hpi. ChIP- and input-DNA samples were quantified by quantitative PCR assays. Relative accumulation levels were represented by percentage of input. **d** Sequential ChIP-qPCR measurements of H3K4me3-H3K27me3 bivalent modification at *BCG1* gene in PH-1 and the indicated mutants at 0–120 hpi. Sequential ChIP was performed by first ChIP using anti-H3K4me3 antibody followed by a second round of ChIP using anti-H3K27me3 antibody. Relative accumulation levels were represented by percentage of input. **e** ChIP-qPCR assays revealed the enrichments of Kmt6-GFP, Set1-GFP, and BP1-GFP at *BCG1* gene at 0–120 hpi. ChIP- and input-DNA samples were quantified

by quantitative PCR assays. **f** Cross-sections of inoculated and adjacent wheat spikelets inoculated with PH-1, and the mutants strains bearing FgActin-RFP. The samples were taken at 7 dpi. Abbreviations: BF, bright field; RFP, red fluorescent protein. **g** The extracellular xylanase activities are determined on the total proteins from the supernatants of PH-1, *ΔSet1*, and double deletion mutant *ΔSet1-BCG1* cultured in 1.1% CMC medium. **h** Quantification of $H_2O_2$ production in wheat seedling leaves at 48 hpi with indicated strains. In (**b**–**e**, **g**, **h**), data presented are the means ± standard deviation ($n = 3$, biologically independent experiments). In (**g**, **h**), statistical significance was assessed using a one-way ANOVA with Fisher's LSD test ($p$ value < 0.01). **i** DCFH-DA- (green fluorescent) and DAB- (dark brown color) staining showing ROS ($H_2O_2$) accumulation in wheat seedling leaves. Bar = 50 μm. **j** Model of epigenetic histone modification-dependent strategy evolved in *F. graminearum* that facilitates successful infection and host immunity evasion during different infection stages. Figure created using BioRender (http://biorender.com/). Source data are provided as a Source Data file.

penetrate host epidermal cell wall within 24 hpi (Fig. 6a and Supplementary Fig. 13). By 48 hpi, Fg successfully broke through the cell wall barrier and intracellularly invaded the host cell (Fig. 6a and Supplementary Fig. 13). We then monitored the *BCG1* expression at these infection stages to understand the contradictory functions of BCG1 in facilitating Fg infection and triggering plant immunity. RT-PCR and RT-qPCR assays showed that *BCG1* expression was highly induced at 24 hpi and followed by a rapid decline after 48 hpi (Fig. 6b and Supplementary Fig. 14a). Protein accumulations of BCG1 during host invasion were further determined using *ΔBCG1::BCG1-Flag* strain. Consistent with RNA levels, immunoblot analysis revealed that BCG1 was highly

accumulated at 24 hpi but abolished after 48 hpi (Supplementary Fig. 14b). Therefore, we hypothesized that Fg secrets BCG1 at early stages of infection to degrade the host cell wall, and rapidly limits its expression to avoid BCG1 accumulation and subsequent host immune response activation after overcoming the plant cell wall barrier (Fig. 6a).

We next investigated the epigenetic dynamics of *BCG1* during host invasion to examine the underlying mechanisms governing the tight transcriptional regulation of *BCG1* and to identify the epigenetic features associated with the activation and silencing of the *BCG1* gene. ChIP-qPCR assays showed markedly elevated H3K4me3 level and weak

H3K27me3 at *BCG1* locus at 24 hpi as compared to those at 0 hpi (Fig. 6c). This observation was consistent with the induction of *BCG1* gene during early infection. Furthermore, we observed newly formed H3K27me3 domains and constant H3K4me3 modification on *BCG1* at 48–120 hpi (Fig. 6c), implying the occurrence of bivalent chromatin modification during biotrophic infection stage of Fg. To verify this hypothesis, sequential ChIP-qPCR assays were performed, which confirmed high H3K4me3-H3K27me3 bivalent signals on the *BCG1* gene at 48–120 hpi, which were correlated with the transcriptional repression of *BCG1* (Fig. 6d). Therefore, we proposed that the bivalent H3K4me3-H3K27me3 modification on *BCG1* may enable its rapid epigenetic silencing, conferring transcriptional plasticity of *BCG1* during infection.

Previous studies revealed that H3K4me3 and H3K27me3 modifications in Fg were respectively catalyzed by the conserved histone methyltransferases Set1 and Kmt6 in Fg[11,14]. We, therefore, performed ChIP-qPCR assays to determine the occupancy of Set1 or Kmt6 at *BCG1* gene loci during infection using PH-1::*Set1-GFP* and PH-1::*Kmt6-GFP* strains. Results showed that Set1-GFP was highly enriched on the *BCG1* gene at 24–120 hpi, while co-occupancy of Set1-GFP and Kmt6-GFP was mainly observed at 48–120 hpi (Fig. 6e), which coincides with bivalent epigenetic marks establishment. These findings indicate that histone methyltransferases Set1 and Kmt6 are recruited to the *BCG1* locus to establish its bivalent histone modification. Next, we generated the deletion mutants of *ΔSet1* and *ΔKmt6*, which exhibited significantly reduced virulence and absence of H3K4me3 and H3K27me3 modifications, respectively (Fig. 6f and Supplementary Fig. 14c, d). Correspondingly, ChIP-qPCR assays revealed that the deposition of H3K4me3 and H3K27me3 at *BCG1* gene was dismissed in *ΔSet1* and *ΔKmt6* mutants, respectively (Fig. 6c). Moreover, we detected that the bivalent signals on *BCG1* were also compromised in the absence of *Set1* or *Kmt6* (Fig. 6d), suggesting the importance of these two histone methyltransferases in establishing bivalent modification on BCG1. Next, we examine the roles of Set1 and Kmt6 in the transcriptional reprogramming of *BCG1*, and our results suggest that the transcriptional activation and protein accumulation of BCG1 at 24 hpi were disrupted in *ΔSet1* mutant (Fig. 6b and Supplementary Fig. 14a, b). As a result, a significant reduction of extracellular xylanase activity was detected in *ΔSet1* and the double deletion mutant *ΔSet1-BCG1* (Fig. 6g), which further confirmed that H3K4me3 modification drives the induction of *BCG1* during early infection stages. Moreover, we also found that the silencing of *BCG1* at 48–120 hpi was suppressed in the absence of *Kmt6* (Fig. 6b and Supplementary Fig. 14a, b). In comparison to PH-1, the *ΔKmt6* mutant also induced a remarkably increased ROS ($H_2O_2$) accumulation in wheat seeding leaves; however, this plant defense activation was strongly compromised in *ΔKmt6-BCG1* double mutant (Fig. 6h, i). Based on these findings, we concluded that histone methyltransferases Set1 and Kmt6 are recruited to the *BCG1* loci to mediate its H3K4me3-H3K27me3 bivalent modification, which drives the rapid epigenetic silencing of *BCG1* and confers host immune evasion.

### BP1 is required for the recognition of H3K27me3 of bivalent modification and repression of *BCG1*

Next, we aimed to elucidate the molecular mechanism of bivalent histone modifications in mediating *BCG1* repression. In plants, the BAH-PHD (BP) proteins function as histone readers and are involved in the recognition of both H3K4me3 and H3K27me3 marks[38,39]. We, therefore, generated the deletion mutants of the only two BP orthologs in Fg and found that *BCG1* silencing was abolished in *ΔBP1*; however, no such effect was observed in *ΔBP2* (Fig. 6b, Supplementary Fig. 14a, b, and Supplementary Fig. 15). *ΔBP1* showed decreased pathogenicity and caused increased ROS accumulation in wheat plants; however, this ROS induction was also dismissed in *ΔBP1-BCG1* mutant (Fig. 6f, h, i and Supplementary Fig. 14d). In contrast to the

findings of BP proteins in plants, BP1 protein in fungi does not alter the trimethylation of H3K4 and H3K27 but mainly binds to the H3K27me3 to enhance the nucleosome residence and to enable the transcriptional repression of H3K27me3-marked genomic regions[27,40]. In line, here we also found that BP1 had no effect on the H3K4me3, H3K27me3, and bivalent enrichments at the *BCG1* gene during infection (Fig. 6c, d). Further, we investigated the dynamics of BP1 occupancy on the *BCG1* gene during host invasion using *ΔBP1::BP1-GFP* strain. Our results indicate that BP1-GFP was strongly deposited at the *BCG1* locus at 48–120 hpi, and to a much lesser extent at 24 hpi (Fig. 6e), which coincides with the establishment of bivalent modification on the *BCG1* gene. Based on these results, we speculated that BP1 mediates the rapid transcriptional repression of *BCG1* by specifically recognizing the H3K27me3 and not the H3K4me3 of the bivalent modification.

## Discussion

Plants and pathogens are engaged in a continuous coevolutionary struggle for survival. In this study, we provided genetic and biochemical evidence that Fg independently evolves a lineage-specific G/Q-rich motif that confers higher xylanase activity of BCG1 for successful invasion; however, the host plants, in turn, specifically recognize this motif to initiate plant defense. Therefore, to successfully breach the host cell wall together with evading the host immunity induced by accumulated BCG1, Fg has evolved a sophisticated mechanism to tightly control the *BCG1* expression. Here, we have shown that Fg utilizes H3K4me3 modification to activate *BCG1* expression during early infection stages and subsequently silence its expression through rapid switching to a bivalent chromatin state (Fig. 6a, j and Supplementary Fig. 16). Thus, our study illustrates a bivalent epigenetic modification-based fine-tuning of *BCG1* expression temporally at a different phase of infection that not only helps in establishment of infection by degrading the host cell wall, but also facilitates host immune escape. Our findings provided a virulence strategy evolved in fungal pathogens during the arms race with hosts, which provides key insights into epigenetic regulation of fungus-host interactions.

Bivalent chromatin modification, characterized by the simultaneous presence of H3K4me3 and H3K27me3 on the same nucleosome, has emerged as an important epigenetic mechanism in both animals and plants[15,17–19]. However, the presence and biological function of such modifications in fungi remain unclear. In this study, we mapped the genome-wide landscape of H3K4me3 and H3K27me3 dynamics and identified a group of genes (*BCGs*) that show increased H3K4me3-H3K27me3 bivalent modification in Fg during its early infection stages on wheat. Among the identified *BCGs*, *BCG1* showed the highest bivalent signals during infection and the strongest impact on Fg pathogenesis, which was thus used for further characterization. Our results revealed that the bivalent histone modification-based epigenetic silencing of *BCG1* enables Fg to evade host immunity, thereby establishing a successful infection. This is the first report illustrating the biological function of bivalent modification in fungi, which represents an important epigenetic mechanism that modulates fungal pathogenesis via fine-tuning their gene expression. Moreover, as both H3K4me3 and H3K27me3 have also been detected in different fungal species[11–14], it can be speculated that bivalent chromatin modification-based host immune evasion is potentially conserved in different fungal pathogens, and efforts should be made in the future to identify bivalent domains and related features in other fungal pathosystems.

Bivalency mainly occurs at promoters of developmental genes and is considered as a key feature of germline and embryonic stem cells, which poises these genes for activation or repression in response to different stimuli, thus guiding mammalian development[15,16,41]. Bivalent promoters were also found to maintain epigenetic plasticity in cancer stem cells[42]. For example, bivalent chromatin configuration at the ZEB1 promoter enables breast cancer cells to respond readily to

microenvironmental signals and enhances tumorigenicity[43]. In the case of plants, bivalent domains were reported to play key roles in the transcriptional regulation of the stress-responsive genes[17–19]. In contrast to mouse ESCs and animal ESCs, chromatin bivalency in *Arabidopsis* sperm cells was also found at gene body regions of Cold Bivalent Genes (CBGs), which were lowly expressed and were potentially involved in cell wall synthesis and stimulus responses[18]. In this study, we also observed bivalent modifications at both promoter and gene body regions of *BCG1* in Fg. It will be interesting in the future to test whether this bivalency feature at *BCG1* is associated with its rapid transcriptional repression during Fg infection. Collectively, these findings indicate that although bivalent modification exists and fine-tunes gene expression in different organisms, the biological functions and features of bivalency may be distinct.

Accumulating evidence indicates that bivalent histone modifications confer a "poised" chromatin state that provides gene expression plasticity to control various biological processes, such as development and stress responses, in animals and plants[15,18,19,41]. In this study, we observed a dynamic reprogramming of bivalency in a group of Fg genes during its successful infection on wheat and demonstrated that *BCG1* gene converts from an active H3K4me3 to a bivalent chromatin state by the specific gain of H3K27me3 for rapid epigenetic silencing of *BCG1* to avoid host immune system recognition. Therefore, we propose that bivalent modification-mediated poised chromatin state may also confer epigenetic plasticity to achieve precise and quantitative control of gene expression in fungi, which could be a robust strategy to enable fungal adaptations to host microenvironmental changes.

COMPASS family and Polycomb Repressive Complex 2 (PRC2) have been implicated as central methyltransferase complexes involved in the deposition of H3K4me3 and H3K27me3 deposition, respectively[44]. Bivalency establishment requires the recruitment of specific histone methyltransferases to the target genes. To date, the molecular mechanism for chromatin bivalency establishment has only been described in animal systems. MLL2, a trithorax group type methyltransferase of the COMPASS family, is involved in the recognition of CpG islands and deposition of H3K4me3 marks at bivalent promoters[45,46], while EZH2 of PRC2 complex is responsible for catalyzing H3K27me3 in bivalency[8,47]. In fungi, the MLL2 was lost during evolution, and Set1 represents the sole H3K4me3 methyltransferase[11,48]. Here, we observed that Set1 is recruited to the *BCG1* loci during host invasion to deposit H3K4me3 marks at bivalent chromatin, supporting that Set1 is the methyltransferase responsible for bivalency in Fg. Consistent with the findings in animals, kmt6, a homolog of EZH2 in Fg, was found to be essential for the H3K27me3 deposition in bivalent chromatin. Deletion of *Kmt6* led to the constitutive monovalent H3K4me3 modification of *BCG1* during infection, leading to its transcription activation. In addition, we also found that Δ*Kmt6* deletion muntant induced increased ROS accumulation and displayed dramatic growth reduction, which may result in its significant defect in fungal virulence. Collectively, our results indicate that Set1 and Kmt6 are histone methyltransferases required for bivalency establishment and maintenance in Fg. However, the molecular mechanisms how Set1 and Kmt6 are precisely recruited to *BCG1* locus remains unclear. Previous studies have shown that bivalency resolution requires histone demethylation by lysine-specific demethylases[49–51]. Chromatin remodeler Asf1a also mediates disassembly at bivalent gene promoters during mouse ESCs differentiation[52]. Future work should clarify whether these lysine-specific demethylases and chromatin remodelers also influence resolution of bivalent chromatin domains in Fg.

In plants, BP proteins, EBS and SHL, recognize both H3K4me3 and H3K27me3 to regulate floral phase transition[38,39]. However, fungal BP protein, BP1/EPR-1, is majorly involved in the recognition of H3K27me3 to increase nucleosome residence and subsequently confer transcriptional repression[27,40]. Consistently, our results also demonstrate

that BP1 preferentially recognizes the H3K27me3 of bivalent modification and thereby reinforces the transcriptional repression of *BCG1*, which may explain the repressive role of bivalency on gene expression.

The plant cell wall is the frontline of the plant defense system[53]. Therefore, microbial pathogens have evolved an arsenal of cell wall-degrading enzymes (CWDEs) to degrade plant cell wall polymers that not only facilitate their entry into the plant cells but also provide nutrition to the invading pathogens[54]. Xylan, a major component of primary cell walls of monocots, such as wheat and maize, can be hydrolyzed by xylanases derived from plant pathogens. Xylanase mainly belongs to GH10 and GH11 protein families and has been implicated as crucial virulence factors in different phytopathogens[55,56]. A substantial number of investigations have shown increased expression of xylanases in phytopathogens during plant infection. For instance, *xyn11A*, a GH11 family xylanase, was highly expressed *in planta* and was required for virulence of *Botrytis cinerea*[57]. Similarly, *SsXyl1* of *Sclerotinia sclerotiorum* was also transcriptionally activated when *A. thaliana* to facilitate plant infection[29]. However, the exact role of these proteins and the underlying mechanism of their functions during plant-pathogen interaction remain unclear. Here, we observed that BCG1 exhibited the highest xylanase activity among the eight GH10 and GH11 family xylanases and is the major xylanase required for full virulence of Fg. *BCG1* was strongly induced with abundant H3K4me3 epigenetic marks during the early infection stage of Fg. Additionally, we found that the H3K4me3 methyltransferase Set1 was recruited to the *BCG1* locus to enable its H3K4me3 modification. Inhibition of *Set1* repressed the expression of *BCG1*, leading to a dramatic decline in Fg xylanase activity. H3K4me3 is a mark associated with an open chromatin state. Thus, we also performed micrococcal nuclease (MNase)-qPCR assays to study chromatin accessibility and nucleosome occupancy on *BCG1* genomic regions. The results showed that nucleosome occupancy at *BCG1* was significantly reduced at 24 hpi as compared with that at 0 hpi (Supplementary Fig. 17), indicating that greater chromatin accessibility on H3K4me3-marked *BCG1* may be positively linked with its transcriptional activation during early infection stage. Collectively, these data indicate that *BCG1* is highly induced via H3K4me3 modification, which facilitates the initial invasion of host tissues. However, additional studies are required to further unravel the complicated regulatory network of the epigenetic induction of BCG1.

Plants detect the presence of microbial pathogens through the recognition of PAMPs by plant plasma membrane-localized PRRs. Since PRRs activation often requires PAMP levels to be above a certain threshold, downregulating PAMP levels is an effective strategy employed by both fungal and bacterial pathogens to avoid host detection[24,58]. For instance, *Pseudomonas syringae* is able to downregulate flagellin biosynthesis during infection to evade FLS2-mediated immune responses[59]. During the biotrophic phase of infection, the fungal maize pathogen *Colletotrichum graminicola* also limits the expression of key β-1,6-glucan synthesis genes, *KRE5* and *KRE6*, to attenuate PTI in host plants[60]. However, the molecular mechanism through which microbial plant pathogens suppress the PAMP biosynthesis to evade PTI is rarely reported. In this study, we showed that BCG1 functions as a PAMP in Fg that triggers the PRRs-based plant immunity. We further showed that there is also a requirement for a certain threshold of BCG1 concentration to evoke plant immunity and that after crossing the host cell wall, secreted BCG1 protein accumulates to exceed this specific threshold, leading to the activation of host immunity. To counteract this plant defense, Fg therefore downregulates *BCG1* expression through bivalent epigenetic modification to avoid extracellular PAMP perception in the host plant. Collectively, our study illustrates a counter-defense strategy evolved in fungal pathogens that epigenetically modulate PAMP gene expression to evade host recognition.

In this study, we have shown that high levels of BCG1 proteins are efficient elicitors in the control of FHB by exogenous spray in field

trials, indicating the potential applications of BCG1 in plant disease control. Our results also revealed that BCG1 contains a previously uncharacterized C-terminal G/Q-rich motif which is responsible for its PAMP function. G/Q-rich motif is a *Fusarium*-specific domain and is functionally conserved across *Fusarium* members. Notably, the fusion of G/Q-rich motif to BCG1 orthologs (GH11 family proteins) in *M. oryzae* and *A. flavus*, lacking a G/Q-rich motif, dramatically increased their PAMP activity, indicating that the G/Q-rich motif could function as a potentiator for promoting the PAMP activity of GH11 family proteins. Therefore, we suggest that the G/Q-rich motif can be deployed for developing novel engineered xylanases as elicitors for agricultural applications. In addition, since the G/Q-rich motif contributed to fungal virulence and showed lineage specificity in *Fusarium* genus, designing small-molecule modulators targeting G/Q-rich motif might provide novel avenues for *Fusarium* disease therapy.

## Methods

### Fungal strains and growth assays

The Fg wild-type strain PH-1 (NRRL 31084) was used as a parental strain for transformation throughout this study. PH-1 and its derivative strains were cultured at 25 °C on potato dextrose agar (PDA) or Fusarium minimal medium (FMM, 0.5 g KCl, 2 g NaNO$_3$, 1 g KH$_2$PO$_4$, 0.5 g MgSO$_4$·7H$_2$O, 30 g sucrose, 200 μL trace elements for 1 L) amended with various compounds for mycelial growth assays as described in the figure legends.

### Mutant generation and complementation

Targeted gene replacement with hygromycin resistance cassette was performed by polyethylene glycol (PEG) mediated protoplast transformation method as previously described[61]. Primers used for amplifying the flanking sequences of each gene are listed in Supplementary Table 1. The resulting deletion mutants (11 *BCGs* mutants, GH11 and GH10 family members mutants, *ΔBP1*, *ΔBP2*, *ΔSet1-BCG1*, and *ΔKmt6-BCG1*) were verified by PCR assays (Supplementary Fig. 2), and for each gene, at least three independent gene replacement mutants with similar phenotypes were identified. All of the knockout mutants were further characterized for defects in wheat infection and vegetative growth (Supplementary Fig. 18). Moreover, the *BCG1* deletion mutant was confirmed by a southern blot assay (Supplementary Fig. 3).

For complementation of mutants, the full-length *BCG1* fragment with its native promoter sequence was cotransformed with *Xho*I-digested *pYF11* into the yeast strain XK1-25 by the yeast gap repair approach[62]. The resulting *pYF11::BCG1* construct was transformed into the protoplast of *ΔBCG1* mutant. Geneticin-resistant transformants so obtained were verified by PCR assays with relevant primers and further confirmed via southern blot assays (Supplementary Fig. 3). Using a similar strategy, catalytic-site mutated (*BCG1^EI18A* and *BCG1^E209A*), *BCG1* lacking G/Q-rich motif (*BCG1^ΔGQ*) or BCG1 lacking the signal peptide (*BCG1^ΔSP*) cassettes were constructed. These Complemented strains were acheived using geneticin (*NEO*) as the second selectable marker. were constructed. All of the deletion mutants and complementation strains obtained in this study were preserved in 15% glycerol at −80 °C.

### Pathogenicity assays

Virulence of each Fg strain on flowering wheat (*Triticum aestivum*) heads and seedling leaves was assayed as described previously[61]. A 10 μL aliquot of fresh conidial suspension (10$^5$ conidia/ml) of each strain was injected into a floret in the middle spikelet of the flowering wheat head (cultivar Jimai22). The control wheat heads were inoculated with 10 μL of sterilized water. Spikelets with typical wheat scab symptoms were examined at 14 days post-inoculation (dpi) to calculate the disease index. The mean ± standard deviation of the disease index was calculated with data from three biological replicates with at least ten wheat heads examined in each replicate.

To assay Fg infection on wheat seedling leaf, a 5-mm mycelial plug collected from the edge of 3-day-old colony was inoculated on a 14-day-old seedling leaf, and the inoculated leaves were incubated at 25 °C and 100% humidity with 12 h of daylight. Data were presented as the mean ± standard deviation from three biological replicates with at least ten seedling leaves examined in each replicate showing the lesion area.

### Protein expression and purification

The full-length ORF of *BCG1* was amplified using the genomic cDNA of PH-1, and then cloned into the *pPICZα* vector[63]. The resulting *pPICZα::BCG1* construct was transformed into *Pichia pastoris* X-33 (Muts$^+$), which was cultured overnight in the YEPD (yeast extract-peptone-dextrose) medium at 30 °C. For protein expression, *P. pastoris* transformants were grown in the BMGY (Buffered Glycerol Complex Medium) for 24 h and then transferred to BMMY (Buffered Methanol-Complex Medium) (pH = 6.5). The recombinant protein BCG1-His was purified from the culture supernatant of *P. pastoris* using HisTrap FF Ni Sepharose Columns (GE Healthcare). The expression and purification of 6 × His tagged BCG1^EI18A, BCG1^E209A, BCG1^ΔGQ, XYLA, XYLB, AfBCG1, and MoBCG1 recombinant proteins were similarly performed as described above.

### Xylanase activity assays

To determine the extracellular xylanase activity of wild-type PH-1 and the mutants, each Fg strain was incubated in 10 mL 1.1% CMC medium (11 g CMC-Na, 2 g (NH$_4$)$_2$SO$_4$, 0.5 g KH$_2$PO$_4$, 2 g K$_2$HPO$_4$, 0.1 g CaCl$_2$, and 0.1 g MgSO$_4$·7H$_2$O for 1 L) for 120 h at 25°C[64]. The culture supernatant was collected and used for detecting the extracellular enzymatic activity of each Fg strain. The xylanase activity of the culture supernatant and purified proteins was further determined as described in Moscetti et al. by a 2,4-dinitrosalicyclic acid assay[65].

### ROS accumulation and callose deposition assays

For investigating the cellular ROS accumulation in wheat plants in response to Fg attack, the wheat seedling leaves were inoculated with PH-1, overexpressed strains (*OE-BCG1*, *OE-BCG1^ΔGQ*, *OE-BCG1^EI18A*, and *OE-BCG1^ΔGQ/EI18A* strains), *ΔKmt6*, and *ΔKmt6-BCG1* for 48 h. The cellular ROS accumulation was visualized via the DCFH-DA and DAB staining as reported previously[66]. Briefly, thin epidermal layers of infected wheat seedling leaves were immersed in 5 μM DCFH-DA (15204, Sigma), and then incubated in the dark for 30 min on a horizontal shaker at 25 °C. DCFH-DA signals were observed under a fluorescence microscope at the excitation/emission wavelengths of 488/525 nm. For DAB staining, wheat seeding leaves were inoculated with DAB solution (1 mg/ml, pH 3.8) at room temperature for 8 h and destained with decolorant solution (ethanol:acetic acid = 94: 4, v/v) for 4 h. ROS localization was visualized with a light microscope. H$_2$O$_2$ content was determined according to the protocol of hydrogen peroxide assay kit (S0038; Beyotime Institute of Biotechnology, Nantong, China). In brief, the reaction mixture containing 50 μL of supernatants and 100 μL of test solution was incubated at 25 °C for 25 min, and the absorbance was measured instantly at a wavelength of 565 nm. Absorbance values were normalized to a standard concentration curve to calculate the concentration of H$_2$O$_2$. Callose deposition assays were performed following an aniline blue-based method as previously described. Briefly, the *N. benthamiana* leaves were destained with a mixture of ethanol:acetic acid (3:1, v/v) and then incubated in aniline blue for 12 h in the dark. Callose deposition was then detected under a fluorescence microscope using a UV filter.

ROS burst was monitored with a luminol/peroxidase-based assay as described previously[67]. Briefly, leaf discs from 5-week-old *N. benthamiana* or 4-week-old *Arabidopsis thaliana* were floated overnight in 200 μL of sterile H$_2$O. H$_2$O was replaced with the reaction solution (17 μg/mL luminol and 10 μg/mL peroxidase supplied with sterile H$_2$O,

1 µM purified 6×His tagged BCG1[E118A], BCG1[E209A], BCG1[ΔGQ], XYLA, XYLB, G/Q-rich, AfBCG1, or MoBCG1 recombinant protein). Luminescence was measured through the GLOMAX96 microplate luminometer (Promega, Madison, WI, USA).

## RNA-seq analysis

For in vitro Fg transcriptome analysis, the mycelia for inoculation were generated from $10^5$ conidia by shaking in 100 ml FMM at 150 rpm for 24 h at 25 °C, which represents the time point of 0 hpi. For in planta fungal transcriptome analysis, the wheat seedling leaves infected with Fg were sampled at 24- and 48-h post-inoculation (hpi). To maximize the enrichment of fungal RNAs, only the infected wheat tissues were excised and sampled for RNA-seq based on observation under a microscope. For transcriptome analysis of wheat plants, the wheat seedling leaves were treated with 1 µM of BCG1 protein or PBS (mock), and samples were collected after 24 h. Total RNA was isolated using TRIZOL reagent (TaKaRa, Dalian, China) following the manufacturer's instructions. RNA-seq was conducted using Illumina HiSeq4000 sequencing system. For RNA-seq data analysis, all of the raw reads were filtered by Fastp (version 0.18.0) software to remove adapters and low quality reads (quality score ≤ 20)[68]. Clean reads were aligned to the reference genome of *F. graminearum* strain PH-1 or wheat *Triticum aestivum* cv. Chinese Spring (IWGSC RefSeq v2.1) using Hisat2 (version 1.3.3)[69]. The cross-mapped reads were removed in order not to inflate the differential expression analysis[70]. The *F. graminearum* genome (accession number: GCA_900044135.1) and the corresponding gene annotation files (version 48) were obtained from EnsemblFungi (https://fungi.ensembl.org/Fusarium_graminearum/Info/Index). Genes displaying log2[fold change] ≥ 1.5 or ≤−1.5 and an adjusted *p* value < 0.05 were classified as differentially expressed genes (DEGs) by DESeq2 package[71]. Expression levels for mRNA in each sample were quantified to TPM (Transcripts Per Million mapped reads) using StringTie (version 1.2.0)[72]. For GO analysis in wheat plants, the top21 categories termed "biological process" with a *q*-value less than 0.05 were chosen for visualization.

## Chromatin immunoprecipitation (ChIP)-seq and sequential ChIP-seq

ChIP experiments were performed as described previously with minor modifications[73,74]. Briefly, 5 g of fresh mycelia inoculated in FMM or wheat heads at 48 hpi were collected for nuclei preparation. Chromatin was digested using 15 µL micrococcal nuclease (MNase: NEB M0247S) for 4 min at 37 °C, and 250 µL of 0.5 M EDTA was added to stop digestion immediately. The enzymatic chromatin was centrifuged, and 200 µL of supernatant was used as input and stored at −20 °C. Immunoprecipitation was conducted using the anti-H3K27me3 (Thermofisher, 39155, MA, USA; 1:500 dilution) or anti-H3K4me3 (Abcam, ab8580, Cambridge, UK; 1:500 dilution) antibody together with the protein A agarose beads (sc-2001, Santa Cruz, CA, USA) overnight at 4 °C. The antibody-beads complexes were successively washed with 1 ml low-salt buffer A (50 mM Tris HCl, pH 7.5, 10 mM EDTA, 50 mM NaCl), middle-salt buffer B (50 mM Tris HCl, pH 7.5, 10 mM EDTA, 100 mM NaCl) and high-salt buffer C (50 mM Tris HCl, pH 7.5, 10 mM EDTA, 150 mM NaCl). After washing, 400 µL elution buffer (50 mM NaCl, 20 mM Tris HCl, pH 7.5, 1% SDS, 5 mM EDTA) was used for eluting the DNA at 65 °C for 20 min. The DNA extracted from the H3K4me3 or H3K27me3 ChIP samples was extracted by phenol/chloroform method. As controls, input DNA was recovered by phenol extraction after the enzymatic digestion step. The extracted DNA was then subjected to high-throughput sequencing on the IlluminaHiseq4000 platform by BGI (Shenzhen, China).

For sequential ChIP assay, chromatin digestion and immunoprecipitation with the first antibody in the primary ChIP procedure were carried out following the protocol described above for ChIP. After washing with low-salt buffer-A, middle-salt buffer-B, and high-salt buffer-C, the chromatin was eluted with elution buffer. SDS of the eluted chromatin was removed by HiPPR Detergent Removal Spin Column Kit (Thermofisher, 88305)[75]. The SDS-removed eluted chromatin was then used for sequential ChIP. Similarly, sequential ChIP with the secondary antibody, elution, DNA purification, and sequencing were performed as described above in ChIP. For each sequential ChIP-seq sample, chromatin was immunoprecipitated with anti-H3K4me3 and subsequent anti-H3K27me3 (H3K4me3-H3K27me3), as well as anti-H3K27me3 and subsequent anti-H3K4me3(H3K27me3-H3K4me3), respectively.

## ChIP-seq and sequential ChIP-seq data analysis

Reads of ChIP-seq and sequential ChIP-seq were obtained, and the quality was controlled using SOAPnuke (version 2.1.7)[76]. The reads containing adapters or more than 1% of unknown nucleotides (N) were removed followed by deletion of the reads with more than 40% bases having a quality value lower than 20. Clean reads, thus obtained, were mapped to the reference genome of *F. graminearum* strain PH-1 by Bowtie2 (version 2.4.5) with default parameters[77]. Only uniquely and concordantly mapped reads were used for further analysis. BPM (Bins Per Million mapped reads, similar to the TPM normalization applied on RNA-seq reads) were calculated to normalized reads using bamCompare tool in Deeptools (version 2.4.1) with input DNA as a control[78]. Two biological replicates were included for ChIP-seq and sequential ChIP-seq assays. The resulting bigwig files were then loaded into genome browser IGV (version 2.8.9)[79] and visually analyzed. MACS2 (version 2.1.4)[80] was used to call peaks as described previously[81]. Peaks were used for further analysis only if they were present in two biological replicates. The peaks were annotated using R package ChIPseeker (version 1.26.2)[82].

To identify differentially enriched ChIP peaks between 48 hpi and 0 hpi in ChIP-seq and sequential ChIP-seq data, we used the R package DiffBind (version 3.0.15)[83] as described previously[81]. Those differentially ChIP peaks with log2[fold change] ≥1.5 or ≤ −1.5 were used to identiy genes. Genes were processed for further analyses only if they displayed increased levels of bivalent histone modifications in both sequential ChIP-seq H3K4me3-H3K27me3 and H3K27me3-H3K4me3 data.

## Real-time PCR

For RT-qPCR assay, the concentrations of purified RNA samples were determined with the Nanodrop 8000 spectrophotometer. An aliquot of 5 µg of total RNA was reverse transcribed using Primescript™ first-Strand cDNA synthesis system (TaKaRa, Dalian, China) with an oligo(dT) primer and cDNA was quantified in real-time PCR using TB Green Fast qPCR Mix kit (TaKaRa, Dalian, China). Quantitative PCR reactions were performed in 20 µL using 96-well plates on an ABI QuantStudio™ 5 machine (ThermoFisher Scientific, Waltham, USA) with the following cycling conditions: 95 °C/30 s and then 40 cycles of 95 °C/5 s, 60 °C/15 s. All of the measurements were followed by melting curve analysis. Results were analyzed using QuantStudio™ 5 machine (ThermoFisher Scientific, Waltham, USA). The comparative cycle threshold (CT) method was used for data analysis and the relative fold difference was expressed as $2^{-\Delta\Delta CT}$. According to gerNorm analysis, we selected *FgACTIN* as an internal control for each quantitative real-time PCR analysis. Primer sequences used for RT-qPCR are listed in Supplementary Table 1.

PCR primers for evaluating ChIP assays and sequential ChIP-qPCR were designed to amplify 120–150 base pair fragments from the indicated genomic regions (Supplementary Table 1). ChIP and sequential ChIP DNA or 1:100 dilution of input DNA was used as template, and relative enrichment levels were represented by percentage of input, calculated by $2^{-\Delta Ct}$ ($=2^{-[CT(IP)-CT(Input)]}$). The ChIP-qPCR and sequential ChIP-qPCR assays were repeated independently three times.

## Agrobacterium-mediated transient expression and virus-induced gene silencing (VIGS) assay in N. benthamiana

Agrobacterium-mediated transient expression in N. benthamiana was performed as described previously[84]. To assay BCG1-triggered cell death in N. benthamiana, the coding sequence of BCG1 was cloned into pGR107 and then introduced into the Agrobacterium tumefaciens strain GV3101 by electroporation. After 16 h of culture in liquid LB medium at 28 °C, Agrobacterium cells carrying BCG1were harvested by centrifugation and resuspended in infiltration medium (10 mM MES, 10 mM MgCl₂, 150 µM acetosyringone). The optical densities of cell suspensions in infiltration medium were adjusted to an OD600 of 0.4. The cells were then incubated for 2 h at room temperature and infiltrated into the leaves of 5-week-old N. benthamiana plants. Leaf samples of N. benthamiana were harvested for protein extraction 48 h after infiltration. For VIGS assays, Agrobacterium strains carrying TRV2:GFP or TRV2:BAK1 or TRV2:SOBIR1 vector and carrying TRV1 vector were mixed in a 1:1 ratio and were infiltrated into N. benthamiana seedlings. TRV:GFP was used as a control. The silencing efficiency of the VIGS assay was evaluated by RT-qPCR. Each assay was repeated three times.

## Co-immunoprecipitation (Co-IP) assay

To test the association of BCG1/BCG1^ΔGQ with BAK1 or SORBIR1, these putative interacting proteins were transiently coexpressed in N. benthamiana. Agroinfiltrated N. benthamiana leaves were homogenized in liquid nitrogen, and total protein was extracted by incubating in cell lysis buffer (Beyotime, p0043) containing 0.1% (v/v) protease inhibitor cocktail (P9599; Sigma, St. Louis, MI, USA) for 20 min. The lysates were centrifuged at 14,000 g for 15 min and 200 µl of supernatant was boiled for 5 min as input. For Co-IP, GFP-trap A beads (Chromotek, Hauppauge, NY, USA, gta-20) were added to the supernatant and incubated overnight at 4 °C. The beads were then harvested and washed with TBS buffer thrice, and transient protein expression was detected by immunoblotting using anti-HA tag (M20003M, Abmart, Shanghai, China) and anti-GFP tag (ab32146, Abcam, Cambridge, UK) antibodies. A dilution of 1:1,000 is used for primary antibodies and 1:10,000 for secondary antibodies (anti-Mouse antibody (926-32210, LI-COR, Lincoln, NE, USA); anti-Rabbit antibody (926-32211, LI-COR, Lincoln, NE, USA)).

## MNase-qPCR

The MNase (micrococcal nuclease)-qPCR assay was used to measure nucleosome occupancy as described previously[85]. Briefly, The collected nuclei pellet was digested using 15 µL micrococcal nuclease (MNase: NEB M0247S) for 4 min at 37 °C, and 250 µL of 0.5 M EDTA was added to stop digestion immediately. The same amount of nuclei isolation without MNase treatment was used as an undigested control. The DNA was extracted by phenol/chloroform method centrifuged, and then used for quantitative PCR with multiple pairs of primers spanning the tested region (Supplementary Table 1). The resulting amplicons, having an average size of 100 with 20 bp overlap. The relative nucleosome occupancy was calculated by normalizing signals from the MNase-treated DNA to the corresponding undigested genomic DNA of each sample. At the end, all values were further normalized to that of FgACTIN 100+ loci for each sample[27,85].

## Dot blot

The specificity of the anti-histone antibodies was tested by dot blot (Supplementary Fig. 19). Different amounts of biotin-labeled histone peptide are blotted onto a nitrocellulose membrane. Western blot analyses were then performed using the anti-histone antibodies: anti-H3K27me3 (Thermofisher, 39155, MA, USA) and anti-H3K4me3 (Abcam, ab8580, Cambridge, UK). A dilution of 1:1000 is used for primary antibodies and 1:10,000 for secondary antibodies. The control experiment was probed using Strepavidin-HRP (Thermo Scientific

Pierce, #21130) using a dilution of 1:1000. The histone peptide used include: Biotinylated Monomethyl Histone H3K4 Peptide (Epigentek, #R-1019-100), Biotinylated Dimethyl Histone H3K4 Peptide (Epigentek, #R-1021-100), Biotinylated Trimethyl Histone H3K4 Peptide (Epigentek, #R-1023-100), Biotinylated Monomethyl Histone H3K27 Peptide (Epigentek, #R-1031-100), Biotinylated Dimethyl Histone H3K27 Peptide (Epigentek, #R-1033-100) and Biotinylated Trimethyl Histone H3K27 Peptide (Epigentek, #R-1035-100).

## Bioinformatics analysis

A detailed list of all the parameters of softs used for RNA-seq and ChIP-seq analyses can be found in Supplementary Table 2.

## Statistics and reproducibility

All data were presented as means ± standard deviation. Number of biologically independent samples (n) was indicated in the figure legends. Statistical comparisons were performed using SPSS version 26 (IBM) software with appropriate methods as indicated in the figure legends.

## Reporting summary

Further information on research design is available in the Nature Portfolio Reporting Summary linked to this article.

## Data availability

Relevant data supporting the findings of this study are available in this article and its Supplementary Information files. The ChIP-Seq and RNA-Seq data generated in this study have been deposited in the NCBI BioProject database under the accession code GSE213962. Source data are provided with this paper.

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

## Acknowledgements

We appreciate Prof. Yufeng Wu at Nanjing Agricultural University for their valuable suggestions. This work was financially supported by the National Key R&D Program of China (2022YFD1400102), the Natural Science Foundation for Excellent Youth Scholars of Jiangsu Province, China (BK20200078), National Natural Science Foundation of China (32072364, 31605189).

## Author contributions

Q.G., Y.M.W., X.Z. designed the experiments; Q.G., X.Z., Y.M.W., B.Y., H.Z., Y.J.W., Z.T., Z.W., H.W., G.L., W.S. conducted the experiments; Q.G directed the project; Q.G., X.G., Z.M., K.T., R.G. analyzed data and wrote the manuscript. All authors read and approved the manuscript.

## Competing interests

The authors declare no competing interests.
