## [Peer Review File · Nature Communications]

Temporally-coordinated bivalent histone modifications of BCG1 enable fungal invasion and immune evasionREVIEWER COMMENTS

Reviewer #1 (Remarks to the Author):

In this paper, the authors identified a fungi bivalent histone modification related gene BCG1 and showed the gene is related to fungal invasion and is controlled by an H3K4me3-H3K27me3 switch. The authors firstly identified the gene BCG1 by ChIP analysis of bivalent histone modification genes. Later, the xylanase activity of BCG1 and its role in fungal invasion were extensively analyzed. Finally, the author confirmed the expression of BCG1 is controlled by Set1 and KMT6 mediated H3K4me3 and H3K27me3 deposition. Although the BCG1 is identified by the analysis of bivalent genes, the regulation of BCG1 more depends on the gene expression stimulation by H3K4me3 and gene silencing by H3K27me3. The suggestions are listed below.

1. The GQ motif is more likely to be a stimulator of the xylanase activity. The GQ deletion BCG1 has lower but significant activity. It is not accurate to claim GQ motif is REQUIRED for its activity.
2. The BCG1 breaks the cell wall and is required for the fungal virulence. Why the kmt6 mutation which eliminate H3K27me3 and stimulate BCG1 expression also reduced the fungal virulence. A rational explanation and discussions are required.

Reviewer #2 (Remarks to the Author):

The main topic of this manuscript deals with fungal invasion and the authors focus on analyzing in detail the role of a gene encoding a xylanase during the infection process. The first map regions in the genome containing histone H3K4me3 and H3K27me3 and select a subset where these two marks are located in the same nucleosome at 48hpi using sequential ChIP. They consider these genes as typical of bivalent chromatin.

Then they make a more detailed analysis of one of those genes, BCG1 to show the relevance of BCG1 in successful infection by its cell wall degrading activity, how it works in pathogen-associated patterns, the role of a G/Q motif, the triggering of plant immunity by BCG1 and other aspects related to fungal infection in wheat.

My main question relates to the point of chromatin bivalency defined by the presence of H3K4me3 and H3K27me3 mark in the same locus (nucleosome), which is otherwise only a relatively small part of the work.

- In lines 62-63 authors state that these marks “have been only identified in a few Arabidopsis genomic regions so far and are potentially linked with the dehydration stress process 17”. The reference is misplaced, leaving the speculation outside.
- Fig 1a,b, line 116. What is the statistical support for “significant”? How does the plot of one signal vs the other look like? Do they overlap in the diagonal? Can you distinguish a population predominantly marked with either H3K4me3 or H3K27me3?
- Fig 1 e-f. The location of these two marks in the promoter and/or gene coding sequences is important to define them as a relevant bivalent chromatin. Typical bivalent regions occur in the promoter regions, allowing in that way to poise genes instead of repressing them in the long term. Please explain and expand this discussion.
- Fig 1g. It seems that bar colors in the lower panel of Fig 1g do not seem to match the code.
- Fig 5. Does the gene overexpressed also acquire the same pattern of H3K4me3 and H3K27me3 marks?
- Fig 6j. It is found that levels of H3K4me3 are high at 24hpi and maintained at 48hpi, a time where H3K27me3 also reach high values. They propose that at this stage a bivalent

chromatin is set to repress gene expression. This situation is opposite to a more frequent setting where a gene is initially poised by having the two marks and then, upon a given stimulus, the H3K27me3 is removed to start active transcription rapidly. Is transcription poised at late stages? Does RNA polII remains at the TSS? Another unknown is how expression early after infection is stimulated and how the BCG1 gene is shut off before initial infection. How does chromatin bivalency remain long after infection? How is chromatin brought back to the condition before BCG1 expression initiates again?

- A general concern is that figures are too crowded, small and consequently difficult to visualize. The colors used in several figures (pale blue, pale green) are not easy to identify.

Reviewer #3 (Remarks to the Author):

The article "Temporally-coordinated bivalent modifications of BCG1 enable fungal invasion and immune invasion" covers the very relevant questions of the significance and function of chromatin regions that are covered with both H3K4me3 and H3K27me3 within the same nucleosome, in the phytopathogenic fungus *Fusarium graminearum*.

The work performed is extensive. It combines transcriptomics, ChIP-seq, sequential ChIP assays, in planta assays, imaging assays, the use of various genetically engineered strains... The main message is that the gene BCG1 - decorated with both H3K4me3 and H3K27me3 - encodes a cell wall degrading xylanase involved in fungal infection of the model plant *Nicotiana benthamiana* and on wheat heads. The authors' results are consistent with the dynamic control of BCG1 expression during the process of infection. Even though these results are highly appealing and I fully acknowledge the work put into the study and the skills of the authors, major flaws on controls/validations not performed raise critical red flags.

First, no validation of the antibodies used is proposed. That is especially true for the sequential ChIP (but not only) where it's critical to know that they work the expected way. ENCODE guidelines wisely recommend to check antibodies specificities by two independent methods. For sequential ChIP, do the authors get the same results doing H3K4me3 first and H3K27me3 first? Another sticky point is the mutants used in the study. I couldn't figure how they were all engineered and checked. The materials and methods section is largely insufficient to know what was done, most mutants are not even mentioned. For BCG1, a southern blot is shown next to a simplified locus exchange map for the KO, but that's it. I really have nothing there that confirms all lights are green although it's quite well known that transformations of fungi can lead to unexpected funny things... The materials and methods section is also largely insufficient to cover all the stats and omics analyses that must have been performed to reach those results. What was performed is quite not transparent (incidentally I also question the validity of RPKM and FPKM usage, as heavily discussed in the literature). As a final detail here, I wondered what was 0 hpi. Did the authors mean spores inoculated?

Either the proper controls/validations have been done and the authors need to re-write the paper taking those into account, or they must be performed to make sure the many artifacts that could occur there (and assess the domain of validity of all experiments performed here) were avoided or accounted for. Once these essential requirements are met, a re-written manuscript may be appropriately reviewed. Considering the amount of critical black zones in the presented paper, I cannot recommend it for publication.

Reviewer #4 (Remarks to the Author):

The manuscript “Temporally-coordinated bivalent histone modifications of BCG1 enable fungal invasion and immune evasion” compared the global H3K4me3 and H3K27me3 histone modifications under two conditions: 48 hpi and in vitro and focused the study on bivalent chromatin modification. Using reverse genetics, author confirmed the importance of some genes with bivalent chromatin modification involved in pathogenesis. Able to identify ONE candidate gene, BCG1 (bivalent chromatin-marked gene1), for the follow up study is clever. The biochemical characterization of BCG1 using Pichia expression system enabled in vitro characterization including confirmed enzymatic property of the protein (Xylanase activity) and test the activity residue and importance the N-terminal the G/Q-rich domain.

This study is interesting, but presentation is very coarse and very hard to follow with some mistakes. Some conclusions are not fully supported by the data.

1) The major question regarding the model: authors claimed “Within 24 hpi, *F. graminearum* establishes surface colonization and activates the expression of BCG1 via H3K4me3 modification for wheat cell wall breakdown. After breaching the host cell wall, *F. graminearum* subsequently limits BCG1 expression through H3K4me3-H3K27me3 bivalent histone modifications to avoid host recognition at 48 hpi.” However the identification of BCG1 is based on the overlap with upregulated genes (Figure 1c).

2) Based in Fig2B and C, the Xylanase activity is reduced but still exist with BCG1ΔGQ. So the conclusion G/Q-rich motif of BCG1 is required for its xylanase activity in line 232 is not supported.

3) Figure 3a and c., the infiltrated leaves are surprisely clean. Why not present full leave images, instead of cutting squares.

4) For Extended Data Figure 3a, please add the mutant BCG1ΔGQ Xylanase activity results

5) In the manuscript, sometime Extended Data Figures are used and sometime Supplementary Figures are called

6) Order of Extended Data Figures are not in order

7) Figure legend doesn't support the independence of each figure. For instance, figure 2 never mentioned that “total protein” actually means “total protein from supernatant”

8) Abstract lacks clarity.

“BCG1, which encodes a xylanase containing a novel G/Q-rich motif, possessed the highest bivalent modification”

– how to define the highest, comparing to what under which conditions?

- Is this gene secreted? is it evolutionary conserved?

“tightly regulates BCG1 expression during different infection stages”

– which and how many stages are inspected?

During initial infection stages, *Fg* employs H3K4me3 modification to induce BCG1 expression required for host cell wall degradation.

- what defines initial infection stages

Subsequently, upon breaching the cell wall barrier, this active chromatin state was reset to bivalency by co-modifying with H3K27me3, which enables epigenetic silencing of BCG1 to escape from host immune surveillance.

- What defines "subsequently"?

Response to referees

Reviewer #1 (Remarks to the Author):

In this paper, the authors identified a fungi bivalent histone modification related gene BCG1 and showed the gene is related to fungal invasion and is controlled by an H3K4me3-H3K27me3 switch. The authors firstly identified the gene BCG1 by ChIP analysis of bivalent histone modification genes. Later, the xylanase activity of BCG1 and its role in fungal invasion were extensively analyzed. Finally, the author confirmed the expression of BCG1 is controlled by Set1 and KMT6 mediated H3K4me3 and H3K27me3 deposition. Although the BCG1 is identified by the analysis of bivalent genes, the regulation of BCG1 more depends on the gene expression stimulation by H3K4me3 and gene silencing by H3K27me3.

Our response: Thank you very much for the comments. Based on your suggestions, we have made changes in the revised manuscript.

1. The GQ motif is more likely to be a stimulator of the xylanase activity. The GQ deletion BCG1 has lower but significant activity. It is not accurate to claim GQ motif is REQUIRED for its activity.

Our response: Thanks for your constructive comments. We do agree with the reviewer that G/Q-rich motif of BCG1 is a stimulator of its xylanase activity. We have revised it throughout the manuscript. Please see lines 32-33, 238-239.

2. The BCG1 breaks the cell wall and is required for the fungal virulence. Why the *kmt6* mutation which eliminate H3K27me3 and stimulate BCG1 expression also reduced the fungal virulence. A rational explanation and discussions are required.

Our response: Thanks for this comment. Although BCG1 is required for cell wall breakdown and serves *F. graminearum* as a virulence factor, high levels of BCG1 also function as an immune elicitor to evoke plant immunity after crossing the host cell wall. In our study, we found that deletion of *Kmt6* leads to the constitutive transcriptional activation of *BCG1* during infection (revised manuscript, Fig 6b), which resulted in activation of host immunity (revised manuscript, Fig. 6h, i). This

may lead to the defects of fungal virulence in $\Delta Kmt6$. Moreover, consistent with previous studies^(1,2), we found that disruptive mutant of *Kmt6* in *F. graminearum* caused dramatic growth reduction (see below Fig. 1), which may also lead to the reduced fungal virulence of $\Delta Kmt6$. As suggested, we discussed this in the revised manuscript (lines 535-539).

Fig. 1 The wild-type strain PH-1 and the $\Delta Kmt6$ mutant were grown on PDA at 25 °C for 3 days.

REFERENCES

- (1) Connolly, L. R., Smith, K. M. & Freitag, M. The *Fusarium graminearum* histone H3 K27 methyltransferase KMT6 regulates development and expression of secondary metabolite gene clusters. *PLoS Genet.* **9**, e1003916 (2013).
- (2) Tang, G., Yuan, J., Wang, J., Zhang, Y. Z., Xie, S. S., Wang, H., Tao, Z., Liu, H., Kistler, H. C., Zhao, Y., Duan, C. G., Liu, W., Ma, Z. & Chen, Y. *Fusarium* BP1 is a reader of H3K27 methylation. *Nucleic Acids Res.* **49**, 10448-10464 (2021).

Reviewer #2 (Remarks to the Author):

The main topic of this manuscript deals with fungal invasion and the authors focus on analyzing in detail the role of a gene encoding a xylanase during the infection process. The first map regions in the genome containing histone H3K4me3 and H3K27me3 and select a subset where these two marks are located in the same nucleosome at 48hpi using sequential ChIP. They consider these genes as typical of bivalent chromatin. Then they make a more detailed analysis of one of those genes, BCG1 to show the relevance of BCG1 in successful infection by its cell wall degrading activity,

how it works in pathogen-associated patterns, the role of a G/Q motif, the triggering of plant immunity by BCG1 and other aspects related to fungal infection in wheat.

My main question relates to the point of chromatin bivalency defined by the presence of H3K4me3 and H3K27me3 mark in the same locus (nucleosome), which is otherwise only a relatively small part of the work.

Our response: We appreciate your professional comments and suggestions that help a lot with improving our manuscript.

- In lines 62-63 authors state that these marks “have been only identified in a few Arabidopsis genomic regions so far and are potentially linked with the dehydration stress process 17”. The reference is misplaced, leaving the speculation outside.

Our response: Thanks for pointing out this. We revised as suggested. Please see lines 63-66 in the revised manuscript.

- Fig 1a,b, line 116. What is the statistical support for “significant”? How does the plot of one signal vs the other look like? Do they overlap in the diagonal? Can you distinguish a population predominantly marked with either H3K4me3 or H3K27me3?

Our response: We appreciate this comment. As reviewer’s suggestion, we found the significant overlapping of modified regions with H3K4me3 and H3K27me3 based on binomial test (p -value = $3.1e^{-15}$). The manuscript was revised accordingly (Line 121). We also calculated and plotted the signal (represented by RPKM) of H3K4me3 and H3K27me3 respectively, for the regions under three categories: H3K4me3 only, H3K27me3 only, H3K4me3-H3K27me3 co-marked (see below Fig. 2). We found the plots of the different categories did not overlap in the diagonal and formed three obvious domains, indicating that it was distinguishable for regions predominantly modified with H3K4me3 only or H3K27me3 only, as well as regions co-marked with both the modifications.

Fig.2 Scatter plot showing H3K4me3 and H3K27me3 levels of regions under three different categories: H3K4me3 only, H3K27me3 only, H3K4me3-H3K27me3 co-marked.

- Fig 1 e-f. The location of these two marks in the promoter and/or gene coding sequences is important to define them as a relevant bivalent chromatin. Typical bivalent regions occur in the promoter regions, allowing in that way to poise genes instead of repressing them in the long term. Please explain and expand this discussion.

Our response: We appreciate the reviewer's comments on this. Bivalency occurs in the promoter regions of developmental genes in both mouse and mammalian embryonic stem cells (ESCs)⁽¹⁾. However, a recent study in *Arabidopsis* sperm cells revealed that, in contrast to mouse and animal ESCs, chromatin bivalency was found at gene body regions of Cold Bivalent Genes (CBGs), which were lowly expressed and were potentially involved in cell wall synthesis and stimulus responses⁽²⁾. In our study, we observed bivalent modifications occur at both promoter and gene body regions of *BCG1* in *F. graminearum* (Fg). These findings indicate that although bivalent modifications exist in different organisms, the bivalency features may be distinct. It will be interesting in the future to test whether this bivalency feature at *BCG1* is associated with its rapid transcriptional repression during Fg infection. As suggested, we have discussed this in the revised manuscript (line 500-506).

REFERENCES

(1) Macrae, T. A., Fothergill-Robinson, J. & Ramalho-Santos, M. Regulation, functions and transmission of bivalent chromatin during mammalian development.

Nat. Rev. Mol. Cell Biol. **24**, 6-26, (2023).

(2) Zhu, D., Wen, Y., Yao, W., Zheng, H., Zhou, S., Zhang, Q., Qu, L.-J., Chen, X. & Wu, Z. Distinct chromatin signatures in the *Arabidopsis* male gametophyte. *Nat. Genet.* **55**, 706-720, (2023).

- Fig 1g. It seems that bar colors in the lower panel of Fig 1g do not seem to match the code.

Our response: Thanks for pointing out this. We corrected the Fig. 1f (the original Fig. 1g).

- Fig 5. Does the gene overexpressed also acquire the same pattern of H3K4me3 and H3K27me3 marks?

Our response: Thanks for this comment. We overexpressed *BCG1* in PH-1 under the control of *gpda*, a strong constitutive promoter from *Aspergillus nidulans*⁽¹⁾. ChIP-qPCR assays showed no deposition of H3K4me3 and H3K27me3 at *BCG1* locus in the overexpression strain *OE-BCG1* (see below Fig. 3a). Moreover, we overexpressed *BCG1* in the deletion mutants of $\Delta Set1$ and $\Delta Kmt6$, which exhibited absence of H3K4me3 and H3K27me3 modifications, respectively (revised manuscript, Supplementary Fig. 14c). RT-qPCR assays showed *BCG1* expression in *OE-BCG1* strain was similar with that in the strains $\Delta Set1::OE-BCG1$ and $\Delta Kmt6::OE-BCG1$ (see below Fig. 3b, c). These results indicate that H3K4me3 and H3K27me3 modifications are not required for the overexpression of *BCG1* under *gpda* promoter.

Fig. 3 H3K4me3 and H3K27me3 modifications are not required for the overexpression of *BCG1* under *gpda* promoter.

REFERENCES

(1) Lubertozzi, D. & Keasling, J. D. Marker and promoter effects on heterologous expression in *Aspergillus nidulans*. *Appl. Microbiol. Biotechnol.* **72**, 1014-1023 (2006).

- Fig 6j. It is found that levels of H3K4me3 are high at 24hpi and maintained at 48hpi, a time where H3K27me3 also reach high values. They propose that at this stage a bivalent chromatin is set to repress gene expression. This situation is opposite to a more frequent setting where a gene is initially poised by having the two marks and then, upon a given stimulus, the H3K27me3 is removed to start active transcription rapidly. Is transcription poised at late stages? Does RNA pol II remains at the TSS?

Our response: We thank the reviewer for the comments. Typical bivalent chromatin state is thought to poise important regulatory genes for future activation or repression in mammalian and mouse embryonic stem cells (ESCs)⁽¹⁾. In this study, we found that *BCG1* gene converts from an active H3K4me3 to a bivalent chromatin state by the specific gain of H3K27me3 at 48-120 hpi, which is lowly expressed during this infection stage. Interestingly, we also observed that *BCG1* bivalency is resolved to H3K4me3 monovalent modification by loss of H3K27me3 modification (see below Fig. 4a-c, unpublished data), which is positively linked with transcriptional activation of *BCG1* at late infection stage (at 148 hpi) (see below Fig. 4d, unpublished data). Therefore, we proposed that bivalent modification of *BCG1* at 48-120 hpi is also in a “poised” chromatin state and provides epigenetic plasticity for future activation, which is similar with the findings in mammalian and mouse ESCs. Since bivalency resolution during late infection stage is not our focus in this story, we didn't put this result into current manuscript and we hope this is fine. In future studies, we would like to clarify the contribution of *BCG1* activation at late infection stage (after 148 hpi) to *F. graminearum* virulence and how bivalency of *BCG1* is resolved. We have briefly discussed this in the revised manuscript (Lines 510-513 and 517-520)

Fig. 4 Bivalency is resolved to H3K4me3 monovalent modification on *BCGI* locus at 144 hours post-inoculation (hpi), which is positively linked with its transcriptional activation during late infection stage.

In mammalian and mouse embryonic stem cells (ESCs), bivalent promoters have been found to be occupied by poised RNA polymerase II (RNA pol II)⁽²⁾. As reviewer's suggestion, we carried out a RNA pol II ChIP-qPCR and detected the RNA pol II deposition around the TSS region of *BCGI* at the timing of *BCGI* bivalency establishment (at 48-120 hpi). The results showed that RNA pol II was significantly enriched around the TSS of *BCGI* at 48-120 hpi (see below Fig. 5), indicating a RNA pol II binding event at bivalently marked *BCGI*, which is consistent with the findings in mammalian and mouse ESCs.

Fig. 5 ChIP-qPCR measurements of RNA Pol II deposition at *BCGI* in wild-type PH-1 at 48-120 hpi

REFERENCES:

(1) Macrae, T. A., Fothergill-Robinson, J. & Ramalho-Santos, M. Regulation, functions and transmission of bivalent chromatin during mammalian development. *Nat. Rev. Mol. Cell Biol.* **24**, 6-26, (2023).

(2) Blanco, E., González-Ramírez, M., Alcaine-Colet, A., Aranda, S. & Di Croce, L. The Bivalent Genome: Characterization, Structure, and Regulation. *Trends Genet.* **36**, 118-131 (2020).

- Another unknown is how expression early after infection is stimulated and how the *BCG1* gene is shut off before initial infection.

Our response: Thanks for this comment. Open chromatin with greater accessibility may facilitate the access of regulatory proteins required for gene upregulation⁽¹⁾. Thus, we performed micrococcal nuclease (MNase)-qPCR assays to study chromatin accessibility and nucleosome occupancy on *BCG1* during early *F. graminearum* infection (at 0-24 hpi). The result showed that genomic region of *BCG1* was packaged in a positioned array of nucleosomes at 0 hpi, which may explain the transcriptional shut off of *BCG1* before initial infection. H3K4me3 is a mark associated with an open chromatin state and active gene expression^(2,3). At 24 hpi (early after *F. graminearum* infection), H3K4me3 is highly enriched on *BCG1*, and the positioned nucleosome array at *BCG1* locus was lost with exposure of nucleosome-free region (see below Fig. 6). Consistent with the nucleosome occupancy data, *BCG1* is highly expressed during early infection stage (at 24 hpi). These results indicate that greater chromatin accessibility on H3K4me3-marked *BCG1* may be positively linked with its transcriptional activation during early infection stage. We have added nucleosome occupancy results in the revised manuscript (revised manuscript, Supplementary Fig. 17) and briefly discussed this (lines 575-581).

Fig. 6 Nucleosome occupancy at *BCGI* in PH-1 at 0 and 24 hpi

Nucleosome occupancy at the genomic region of the target gene *BCGI* (-580 to 320 bp) in wild-type PH-1 is determined by MNase-qPCR. The values are the means \pm standard deviation ($n=3$ biological replicates).

REFERENCES:

- (1) Klemm, S. L., Shipony, Z. & Greenleaf, W. J. Chromatin accessibility and the regulatory epigenome. *Nat. Rev. Genet.* **20**, 207–220 (2019).
- (2) Santos-Rosa, H., Schneider, R., Bannister, A. J., Sherriff, J., Bernstein, B. E., Emre, N. C., Schreiber, S. L., Mellor, J. & Kouzarides, T. Active genes are tri-methylated at K4 of histone H3. *Nature* **419**, 407-411 (2002).
- (3) Alvarez-Errico D, Vento-Tormo R, Sieweke M, Ballestar E. Epigenetic control of myeloid cell differentiation, identity and function. *Nat. Rev. Immunol.* **15**, 7–17 (2015).

- How does chromatin bivalency remain long after infection? How is chromatin brought back to the condition before *BCGI* expression initiates again?

Our response: We appreciate the reviewer’s comments. Bivalency establishment and maintenance require precise recruitment of H3K4me3 and H3K27me3 methyltransferases to the target genes⁽¹⁾. In this study, we simultaneously observed high Set1 (H3K4me3 methyltransferase) and Kmt6 (H3K27me3 methyltransferase) occupancies at *BCGI* loci at 48-120 hpi, which may lead to retain of bivalency on

BCG1 during *F. graminearum* infection. However, the underlying mechanism how Set1 and Kmt6 are precisely recruited to *BCG1* locus during *F. graminearum* infection remains unclear, which is over the scope of this manuscript. We have briefly discussed this in the revised manuscript. See lines 539-541.

Previous studies in mammalian and mouse embryonic stem cells (ESCs) showed that chromatin bivalency resolution requires histone demethylation by lysine-specific demethylase 5B (KDM5B; H3K4 demethylation) or by KDM6A and KDM6B (H3K27me3 demethylation)⁽²⁻⁴⁾. In addition, chromatin remodeler, such as Asf1a, also mediates disassembly at bivalent gene promoters during mouse ESCs differentiation⁽⁵⁾. Since bivalency resolution at late infection stage is not our focus in this story, we didn't test whether these histone demethylases or chromatin remodelers function in bivalency resolution in *F. graminearum* during infection. Our future work would focus on the roles of these factors in regulating bivalency resolution in *F. graminearum*. We also briefly discussed this in the revised manuscript. Please see lines 542-547.

REFERENCES:

- (1) Macrae, T. A., Fothergill-Robinson, J. & Ramalho-Santos, M. Regulation, functions and transmission of bivalent chromatin during mammalian development. *Nat. Rev. Mol. Cell Biol.* **24**, 6-26, (2023).
- (2) Kidder, B. L., Hu, G. & Zhao, K. KDM5B focuses H3K4 methylation near promoters and enhancers during embryonic stem cell self-renewal and differentiation. *Genome Biol.* **15**, R32 (2014).
- (3) Dahle, Y., Kumar, A. & Kuehn, M. R. Nodal signaling recruits the histone demethylase Jmjd3 to counteract polycomb-mediated repression at target genes. *Sci. Signal.* **3**, ra48 (2010).
- (4) Dhar, S. S. *et al.* An essential role for UTX in resolution and activation of bivalent promoters. *Nucleic Acids Res.* **44**, 3659–3674 (2016).
- (5) Gao, Y., Gan, H., Lou, Z. & Zhang, Z. Asf1a resolves bivalent chromatin domains for the induction of lineage-specific genes during mouse embryonic stem cell differentiation. *Proc. Natl Acad. Sci. USA* **115**, E6162-e6171 (2018).

- A general concern is that figures are too crowded, small and consequently difficult to visualize. The colors used in several figures (pale blue, pale green) are not easy to identify.

Our response: We thank the reviewer for the comments. As suggested, we have re-organized the figures and put the original Fig. 1a, and Fig. 3c-e into the Supplementary data. Furthermore, we also revised the colors of the figures based on reviewer's suggestion. See Fig. 2b/ Fig. 3d-f/ Fig. 4b/ Fig. 5d, h/ Fig. 6b-e, g, h in the revised manuscript.

Reviewer #3 (Remarks to the Author):

The article "Temporally-coordinated bivalent modifications of BCG1 enable fungal invasion and immune invasion" covers the very relevant questions of the significance and function of chromatin regions that are covered with both H3K4me3 and H3K27me3 within the same nucleosome, in the phytopathogenic fungus *Fusarium graminearum*.

The work performed is extensive. It combines transcriptomics, ChIP-seq, sequential ChIP assays, in planta assays, imaging assays, the use of various genetically engineered strains... The main message is that the gene BCG1 - decorated with both H3K4me3 and H3K27me3 - encodes a cell wall degrading xylanase involved in fungal infection of the model plant *Nicotiana benthamiana* and on wheat heads. The authors' results are consistent with the dynamic control of BCG1 expression during the process of infection. Even though these results are highly appealing and I fully acknowledge the work put into the study and the skills of the authors, major flaws on controls/validations not performed raise critical red flags.

Our response: We appreciate your recognition of our manuscript. Your professional comments and suggestions improve our manuscript greatly.

- First, no validation of the antibodies used is proposed. That is especially true for the sequential ChIP (but not only) where it's critical to know that they work the expected

way. ENCODE guidelines wisely recommend to check antibodies specificities by two independent methods.

Our response: According to ENCODE guidelines, we checked antibodies specificities by dot blot and western blot⁽¹⁾. In western blot assay, we detected specific H3K4me3 and H3K27me3 bands in wild-type strain PH-1 by using the anti-H3K4me3 and H3K27me3 antibodies, but not in the mutants $\Delta Set1$ and $\Delta Kmt6$, respectively (revised manuscript, Supplementary Fig. 14c). Furthermore, the dot blot assay also confirmed the specificities of anti-H3K4me3 and anti-H3K27me3 antibodies. We have added the dot blot results in Supplementary Fig. 18 and described the corresponding information in Materials and Methods section. See lines 859-873.

REFERENCES:

(1) Landt, S.G. *et al.* ChIP-seq guidelines and practices of the ENCODE and modENCODE consortia. *Genome Res.* **22**, 1813–1831 (2012).

- For sequential ChIP, do the authors get the same results doing H3K4me3 first and H3K27me3 first?

Our response: Thanks for this comment. As suggested, for each sequential ChIP-seq sample, chromatin was immunoprecipitated with anti-H3K4me3 and subsequent anti-H3K27me3 (H3K4me3-H3K27me3), as well as anti-H3K27me3 and subsequent anti-H3K4me3 (H3K27me3-H3K4me3), respectively, at 0 and 48 hpi. Bivalent chromatin-marked genes were processed for further analysis only if they displayed increased levels of bivalent histone modifications in both sequential ChIP-seq H3K4me3-H3K27me3 and H3K27me3-H3K4me3 data during *F. graminearum* infection. In the revised manuscript, we identified 11 infection-related *bivalent chromatin-marked genes (BCGs)* (revised manuscript, Fig. 1b-d), which is highly similar to our previous results. Furthermore, we also added the details about bioinformatic analyses for sequential ChIP-seq data in Materials and Methods section (lines 768-770 and 791-793).

- Another sticky point is the mutants used in the study. I couldn't figure how they were all engineered and checked. The materials and methods section is largely insufficient to know what was done, most mutants are not even mentioned. For BCG1, a southern blot is shown next to a simplified locus exchange map for the KO, but that's it. I really have nothing there that confirms all lights are green although it's quite well known that transformations of fungi can lead to unexpected funny things...

Our response: Thanks for this comment. We have added the gene replacement strategy and PCR identification results for all of the mutants (revised manuscript, Supplementary Fig. 2 and 3). For each gene, at least three independent gene replacement mutants were identified. We also added the corresponding information in the Materials and Methods section (lines 632-635 and 643-647).

- The materials and methods section is also largely insufficient to cover all the stats and omics analyses that must have been performed to reach those results. What was performed is quite not transparent (incidentally I also question the validity of RPKM and FPKM usage, as heavily discussed in the literature).

Our response: Thanks for pointing out this. We have revised Materials and Methods section in more detail. See lines 723-728, 732-739, 778-793, 848-857, and 859-873. In our revised manuscript, expression levels for mRNA in each sample were also quantified using TPM (Transcripts Per Million mapped reads). ChIP-seq and sequential ChIP-seq signals were normalized by input using BPM (Bins Per Million mapped reads). As shown in revised Fig. 1c and Supplementary Fig.1d, we got the similar results with our previous results using RPKM and FPKM.

As a final detail here, I wondered what was 0 hpi. Did the authors mean spores inoculated?

Our response: Samples of *F. graminearum* mycelia used for inoculation were collected to represent the time point of 0 hpi. We have added this information in the Materials and Methods section. Please see lines 723-724.

Either the proper controls/validations have been done and the authors need to re-write the paper taking those into account, or they must be performed to make sure the many artifacts that could occur there (and assess the domain of validity of all experiments performed here) were avoided or accounted for. Once these essential requirements are met, a re-written manuscript may be appropriately reviewed. Considering the amount of critical black zones in the presented paper, I cannot recommend it for publication.

Our response: We appreciate reviewer for the valuable comments and suggestions. We hope our reversions have addressed concerns raised by reviewer.

Reviewer #4 (Remarks to the Author):

The manuscript “Temporally-coordinated bivalent histone modifications of BCG1 enable fungal invasion and immune evasion” compared the global H3K4me3 and H3K27me3 histone modifications under two conditions: 48 hpi and in vitro and focused the study on bivalent chromatin modification. Using reverse genetics, author confirmed the importance of some genes with bivalent chromatin modification involved in pathogenesis. Able to identify ONE candidate gene, BCG1 (bivalent chromatin-marked gene1), for the follow up study is clever. The biochemical characterization of BCG1 using Pichia expression system enabled in vitro characterization including confirmed enzymatic property of the protein (Xylanase activity) and test the activity residue and importance the N-terminal the G/Q-rich domain.

This study is interesting, but presentation is very coarse and very hard to follow with some mistakes. Some conclusions are not fully supported by the data.

Our response: Thank you very much for your positive comments on the manuscript. We have revised the manuscript as your suggestions.

1) The major question regarding the model: authors claimed “Within 24 hpi, *F. graminearum* establishes surface colonization and activates the expression of BCG1 via H3K4me3 modification for wheat cell wall breakdown. After breaching the host cell wall, *F. graminearum* subsequently limits BCG1 expression through

H3K4me3-H3K27me3 bivalent histone modifications to avoid host recognition at 48 hpi.” However the identification of BCG1 is based on the overlap with upregulated genes (Figure 1c).

Our response: The identification of *BCG1* is based on the overlap with differential expression genes (DEGs) (48 hpi vs 0 hpi, or 48 hpi vs 24 hpi) and increased bivalent modification genes (48 hpi vs 0 hpi) during *F. graminearum* infection. RNA-seq data revealed that *BCG1* is strongly induced at 24 hpi, but markedly declined at 48 hpi, which undergoes extensive transcriptional reprogramming during *F. graminearum* infection. Thus, we propose BCG1 plays an important role in fungal pathogenesis of *F. graminearum*.

2) Based in Fig2B and C, the Xylanase activity is reduced but still exist with BCG1^{ΔGQ}. So the conclusion G/Q-rich motif of BCG1 is required for its xylanase activity in line 232 is not supported.

Our response: Thanks for your constructive comments. We do agree with the reviewer that G/Q-rich motif of BCG1 is a stimulator of its xylanase activity. We have revised it throughout the manuscript. Please see lines 32-33 and 238-239.

3) Figure 3a and c., the infiltrated leaves are surprisely clean. Why not present full leave images, instead of cutting squares.

Our response: Thank you for your valuable suggestions. We have recreated the Figure 3a and Supplementary Fig. 11b (the original Fig. 3c) in the revised manuscript.

4) For Extended Data Figure 3a, please add the mutant BCG1^{ΔGQ} Xylanase activity results

Our response: Thank you for your helpful suggestion. We have added the mutant BCG1^{ΔGQ} xylanase activity results in Supplementary Fig. 7a (the original Extended Data Figure 3a).

5) In the manuscript, sometime Extended Data Figures are used and sometime

Supplementary Figures are called

6) Order of Extended Data Figures are not in order

Our response: Thanks for pointing out this. We have used Supplementary Figures in the revised manuscript and carefully checked to make all Supplementary Figures in order.

7) Figure legend doesn't support the independence of each figure. For instance, figure 2 never mentioned that "total protein" actually means "total protein from supernatant"

Our response: Thanks for your constructive comments. We have carefully checked the figure legends and revised it. Please see the Fig. 2c, 6g in the revised manuscript.

8) Abstract lacks clarity.

"BCG1, which encodes a xylanase containing a novel G/Q-rich motif, possessed the highest bivalent modification"

– how to define the highest, comparing to what under which conditions?

- Is this gene secreted? is it evolutionary conserved?

Our response: As suggested, we revised it to "*BCG1* gene, which encodes a secreted *Fusarium*-specific xylanase containing a novel G/Q-rich motif, displayed the highest increase of bivalent modification during Fg infection". Please see lines 30-32.

"tightly regulates BCG1 expression during different infection stages"

– which and how many stages are inspected?

Our response: Previous studies have shown that *F. graminearum* (Fg) breaches the host cell wall barrier to initiate its infection. Then, Fg deploys a biotrophic strategy to successfully invades the host cell. Ultimately, Fg causes tissue necrosis, leading to the spreading of pathogen throughout the wheat tissues^(1,2). To avoid misleading, we have deleted this sentence in the revised manuscript.

REFERENCES:

(1) Zhang, X. W., Jia, L. J., Zhang, Y., Jiang, G., Li, X., Zhang, D. & Tang, W. H. In planta stage-specific fungal gene profiling elucidates the molecular strategies of

Fusarium graminearum growing inside wheat coleoptiles. *Plant Cell* **24**, 5159-5176 (2012).

(2) Qiu, H., Zhao, X., Fang, W., Wu, H., Abubakar, Y. S., Lu, G., Wang, Z. & Zheng, W. Spatiotemporal nature of *Fusarium graminearum*–wheat coleoptile interactions. *Phytopathol. Res.* **1**, 26 (2019).

During initial infection stages, Fg employs H3K4me3 modification to induce BCG1 expression required for host cell wall degradation.

- what defines initial infection stages

Our response: Thanks for this comment. *F. graminearum* (Fg) breaches the host cell wall barrier to initiate its infection. We have revised it to “We further discovered that Fg employs H3K4me3 modification to induce *BCG1* expression required for host cell wall breakdown to initiate its early infection”. Lines 35-37.

Subsequently, upon breaching the cell wall barrier, this active chromatin state was reset to bivalency by co-modifying with H3K27me3, which enables epigenetic silencing of BCG1 to escape from host immune surveillance.

- What defines “subsequently”?

Our response: To avoid misleading, we deleted the “subsequently” and revised it to “After breaching the cell wall barrier, this active chromatin state was reset to bivalency by co-modifying with H3K27me3, which enables epigenetic silencing of BCG1 to escape from host immune surveillance. Lines 37-39.

REVIEWER COMMENTS

Reviewer #1 (Remarks to the Author):

All my concerns are properly responded.

Reviewer #2 (Remarks to the Author):

Authors have addressed satisfactorily the questions and corrections listed in my report. They have added new data in the revised version and included changes in the text.

Reviewer #3 (Remarks to the Author):

The authors of the article "Temporally-coordinated bivalent modifications of BCG1 enable fungal invasion and immune invasion" have remarkably improved their manuscript in this revised version. A few questions remain on my side. I really want to be convinced by the results, but I guess I'm still puzzled and confused by some aspects of them; some clarifications are still needed before publication.

- The authors explain 0 hpi is mycelia used for inoculation. I'm very confused, since I assumed (wrongly so) that inoculations in vitro were performed using spores (same as in planta). Can you please explain how inoculum was obtained and normalized between experiments, what is minimal medium? And why did you use mycelium rather than spores? At equal weights, it is so hard to get nicely biologically consistent mycelia from one time to another.
- The materials and methods section still misses important info regarding some of the analyses performed to reach those results, and appreciate the domains of validity for each finding. The authors added a lot of much appreciated info, but I'm missing specificities. 1) shotgun RNA-seq of *Fusarium graminearum* in planta is definitely not trivial, and can sometimes end up looking for a needle in a haystack. How many reads belonged to plant/Fg in each sample? How did you deal with dual mappings (if any)? What distribution of counts did you get for each organism? 2) what annotation of FgPH-1 did you use? 3) how did you make the input control for ChIP, mechanical shearing? Enzymatic? Please, disclose your methods. 4) For ChIP-qPCR you used mock IgG as a control "Control IgG represents the control for the ChIP specificity" (legend fig1). Why not use it also in ChIP-seq? 5) How did you normalize MNase-qPCR assays? I'm not convinced (yet) that what you are showing is true. Accurate quantification is a tricky thing. 6) To ensure data FAIRness, please give parameters of softs you used in a supplemental table for example.
- I must say I'm not very much at ease with the ChIP profiles shown in supp fig 1. The in vitro H3K27me3 is not quite as expected from what is known, i.e., H3K27me3 marks very strong and localized islands in all 4 chromosomes (see Connolly et al 2013). Connolly's pattern has been extensively proved and reproduced in various Fg labs. Any idea on why we do not observe this here? What could be the consequences on the bivalence you propose? Please develop and discuss.
- How did FgACTIN perform as an internal reference for qPCR gene expression analyses? Best practices are to use more than one gene as internal reference and evaluate their validity, using genorm or equivalent for example (and I believe genorm is built in QuantStudio suite, so it should be an easy thing to do to test several housekeeping genes and make sure they behave well).
- For dotblot validations, please provide alignments of sequences in supp data showing that

the peptides you used match the ones found in Fg.

- Is figure 1 H3K4m3 then H3K27me3 or the reverse? Or the intersect? Please clarify.
- The mutants are still a bit of a sticky point to me. Have you checked “some phenotypes” for all 3 mutants you made or did you perform add-backs? Best practices recommend caution (and especially when nearly lethal deletions are considered, such as such as $\Delta set1$ and $\Delta kmt6$ in Fg).

Reviewer #4 (Remarks to the Author):

I am happy with this revision, No further questions.

Response to referees

Reviewer #1 (Remarks to the Author):

All my concerns are properly responded.

Our response: We appreciate the constructive comments made by the reviewer.

Reviewer #2 (Remarks to the Author):

Authors have addressed satisfactorily the questions and corrections listed in my report.

They have added new data in the revised version and included changes in the text.

Our response: We thank the reviewer for the appreciation of our revised manuscript.

Reviewer #3 (Remarks to the Author):

The authors of the article "Temporally-coordinated bivalent modifications of BCG1 enable fungal invasion and immune invasion" have remarkably improved their manuscript in this revised version. A few questions remain on my side. I really want to be convinced by the results, but I guess I'm still puzzled and confused by some aspects of them; some clarifications are still needed before publication.

Our response: We thank the reviewer for the positive evaluation of our revised manuscript. We have provided additional data and revised our manuscript to address the concerns raised by the reviewer.

- The authors explain 0 hpi is mycelia used for inoculation. I'm very confused, since I assumed (wrongly so) that inoculations in vitro were performed using spores (same as in planta). Can you please explain how inoculum was obtained and normalized between experiments?

Our response: We are sorry for missing the details here. For normalization of initial inoculum in the RNA-seq experiments, equal amounts of mycelia were used for inoculation in each biological replicate, which were derived from 10^5 conidia by shaking in 100 ml liquid Fusarium minimal medium (FMM) at 150 rpm for 12 hours at 25°C. We have added the corresponding information in the Methods and Materials section (lines 728-729) of the revised manuscript.

-And why did you use mycelium rather than spores?

Our response: For transcriptomic analysis of filamentous plant pathogens *in planta*, both mycelia and spores can be used as initial inoculum for plant infection⁽¹⁻⁸⁾. During initial *Fusarium graminearum* (Fg) infection, conidia (spores) firstly germinate into runner hyphae and then form various types of penetration structures to successfully invade host tissue⁽⁹⁾. Fg has been found to undergo extensive transcriptional reprogramming during both host infection and conidial germination⁽¹⁰⁾. A recent study has also confirmed significant changes in genes expression of Fg during conidial germination under both *in vitro* and *in planta* conditions⁽¹¹⁾. In this study, we performed a comparative transcriptomic analysis to identify Fg genes involved in fungal pathogenesis. Therefore, to rule out the potential interference from transcriptional reprogramming during conidial germination and discover the infection-specific gene expression alterations, mycelia obtained from germinated conidia were used for wheat inoculation in RNA-seq experiments.

REFERENCES

- (1) Peyraud, R., Mbengue, M., Barbacci, A. & Raffaele, S. Intercellular cooperation in a fungal plant pathogen facilitates host colonization. *Proc. Natl Acad. Sci. USA* **116**, 3193-3201, (2019).
- (2) Wang, Q., Han, C., Ferreira, A. O., Yu, X., Ye, W. et al. Transcriptional programming and functional interactions within the *Phytophthora sojae* RXLR effector repertoire. *Plant Cell* **23**, 2064-2086 (2011).
- (3) Ye, W., Wang, X., Tao, K., Lu, Y., Dai, T., Dong, S., Dou, D., Gijzen, M. & Wang, Y. Digital gene expression profiling of the *Phytophthora sojae* transcriptome. *Mol. Plant Microbe Interact.* **24**, 1530-1539 (2011).
- (4) Soulie, M. C., Koka, S. M., Floch, K., Vancostenoble, B., Barbe, D. et al. Plant nitrogen supply affects the *Botrytis cinerea* infection process and modulates known and novel virulence factors. *Mol. Plant Pathol.* **21**, 1436-1450 (2020).
- (5) Yan, X., Tang, B., Ryder, L. S., MacLean, D., Were, V. M. et al. The transcriptional landscape of plant infection by the rice blast fungus *Magnaporthe*

oryzae reveals distinct families of temporally co-regulated and structurally conserved effectors. *Plant Cell* **35**, 1360-1385 (2023).

(6) Schmidt, S. M., Lukasiewicz, J., Farrer, R., van Dam, P., Bertoldo, C. & Rep, M. Comparative genomics of *Fusarium oxysporum* f. sp. *melonis* reveals the secreted protein recognized by the *Fom-2* resistance gene in melon. *New Phytol.* **209**, 307-318 (2016).

(7) O'Connell, R. J., Thon, M. R., Hacquard, S. et al. Lifestyle transitions in plant pathogenic *Colletotrichum* fungi deciphered by genome and transcriptome analyses. *Nat. Genet.* **44**, 1060-1065 (2012).

(8) Asai, S., Rallapalli, G., Piquerez, S. J., Caillaud, M. C., Furzer, O. J., et al. Expression profiling during arabidopsis/downy mildew interaction reveals a highly-expressed effector that attenuates responses to salicylic acid. *PLoS Pathog.* **10**, e1004443 (2014).

(9) Boenisch, M. J. & Schäfer, W. *Fusarium graminearum* forms mycotoxin producing infection structures on wheat. *BMC Plant Biol.* **11**, 110 (2011).

(10) Brauer, E. K., Subramaniam, R. & Harris, L. J. Regulation and dynamics of gene expression during the life cycle of *Fusarium graminearum*. *Phytopathology.* **110**, 1368-1374 (2020).

(11) Miguel-Rojas, C., Cavinder, B., Townsend, J. P. & Trail, F. Comparative transcriptomics of *Fusarium graminearum* and *Magnaporthe oryzae* spore germination leading up to infection. *mBio.* **14**, e0244222 (2023).

-what is minimal medium?

Our response: Minimal medium (MM) used for culturing *Fusarium graminearum* in this study is actually Fusarium Minimal Medium (FMM). FMM contains 1 g/L KH_2PO_4 , 0.5 g/L $\text{MgSO}_4 \cdot 7\text{H}_2\text{O}$, 0.5 g/L KCl, 2 g/L NaNO_3 , 30 g/L sucrose, as well as 200 $\mu\text{L/L}$ of a trace element solution that was added after autoclaving. FMM used in this study was Puhalla's MM, a sucrose-salt medium that contains nitrate as the nitrogen source⁽¹⁾. FMM is also one of the most commonly used minimal medium for culturing *Fusarium* species⁽²⁻³⁾. We have accordingly corrected 'MM' to 'FMM' and

added the corresponding information in the revised manuscript (lines 627-629).

REFERENCES

- (1) Katan, T., Zamir, D., Sarfatti, M. & Katan, J. Vegetative compatibility groups and subgroups in *Fusarium oxysporum* f. sp. *radicis-lycopersici*. *Phytopathology* **81**, 255-262 (1991).
- (2) Reyes-Dominguez, Y., Boedi, S., Sulyok, M., Wiesenberger, G., Stoppacher, N., Krska, R. & Strauss, J. Heterochromatin influences the secondary metabolite profile in the plant pathogen *Fusarium graminearum*. *Fungal Genet. Biol.* **49**, 39-47 (2012).
- (3) Wang, H., Chen, Y., Hou, T., Jian, Y. & Ma, Z. The very long-chain fatty acid elongase FgElo2 governs tebuconazole sensitivity and virulence in *Fusarium graminearum*. *Environ. Microbiol.* **24**, 5362-5377 (2022).

-At equal weights, it is so hard to get nicely biologically consistent mycelia from one time to another.

Our response: We appreciate the reviewer's comments. In this study, to obtain the biologically consistent inoculum, mycelia used for wheat inoculation were generated from an equal number of conidia shaken under the same cultural conditions. Principal component analysis (PCA) showed that the three biological replicates from each group (at 0, 24, and 48 hpi) were clustered together (please see below Fig. 1), confirming the reproducibility of the biological replicates in the RNA-seq experiments. This result also supports that the inoculum we used in this study is overall biologically consistent.

Fig. 1 Principal component analysis (PCA) plot of all *F. graminearum* genes at 0, 24, and 48 hpi. Three independent biological replicates were generated for each time point 0, 24, and 48 hours post inoculation.

- The materials and methods section still misses important info regarding some of the analyses performed to reach those results, and appreciate the domains of validity for each finding. The authors added a lot of much appreciated info, but I'm missing specificities.

Our response: We appreciate reviewer for the constructive comments. Based on your suggestions, we have revised the Materials and Methods section.

1) shotgun RNA-seq of *Fusarium graminearum* in planta is definitely not trivial, and can sometimes end up looking for a needle in a haystack. How many reads belonged to plant/Fg in each sample? What distribution of counts did you get for each organism?

Our response: Thanks for this comment. We totally agree with the reviewer's comment that it is difficult in the enrichment of fungal RNAs in infected samples during RNA-seq of fungal pathogens *in planta*. Previous studies have revealed that

the removal of uninfected plant tissues can enhance the abundance of fungal RNAs during *in planta* RNA-seq^(1,2). In this study, we used a similar method to maximize the enrichment of fungal RNAs, which only the infected wheat tissues were excised and collected for RNA-seq based on observation under a microscope. By using this approach, the coverage of fungal transcriptome was highly enriched in RNA-seq of *Fusarium graminearum in planta*. Even during early infection (at 24 hpi), we obtained 69895860, 60793486, and 60844626 clean reads for each *in planta* sample, of which 50.57%, 33.49%, and 34.44% were mapped to the *F. graminearum* reference genome, and 47.55%, 64.07%, and 63.21% were mapped to the wheat reference genome. We have added the corresponding information in the Materials and Methods section (Lines 732-734).

REFERENCES

- (1) Lu, X., Kracher, B., Saur, I. M., Bauer, S., Ellwood, S. R., Wise, R., Yaeno, T., Maekawa, T. & Schulze-Lefert, P. Allelic barley MLA immune receptors recognize sequence-unrelated avirulence effectors of the powdery mildew pathogen. *Proc. Natl Acad. Sci. USA*. **113**, E6486-E6495 (2016).
- (2) Jeon, J., Lee, G.W., Kim, K.T., Park, S.Y., Kim, S. et al. Transcriptome profiling of the rice blast fungus *Magnaporthe oryzae* and its host *Oryza sativa* during infection. *Mol. Plant Microbe Interact.* **33**, 141–144 (2019).

-How did you deal with dual mappings (if any)?

Our response: In this study, we identified between 1705 and 2388 cross-mapped clean reads (dual mappings) in each *in planta* sample, which accounted for 0.002% to 0.01% of the total reads. These cross-mapped reads were removed in order not to inflate the differential expression analysis⁽¹⁾. We have added the corresponding information in the Materials and Methods section (Lines 742-743).

REFERENCES

- (1) Westermann, A. J. & Vogel, J. Host-pathogen transcriptomics by dual RNA-Seq. *Methods Mol. Biol.* **1737**, 59-75 (2018).

2) what annotation of FgPH-1 did you use?

Our response: We appreciate the reviewer's comment. The *Fusarium graminearum* genome (accession number: GCA_900044135.1) and the corresponding gene annotation files (version 48), which were obtained from EnsemblFungi (https://fungi.ensembl.org/Fusarium_graminearum/Info/Index), were used in this study. We have added this information in the Materials and Methods section (Lines 743-746).

3) how did you make the input control for ChIP, mechanical shearing? Enzymatic? Please, disclose your methods.

Our response: Thanks for this comment. As controls, input DNA was recovered by phenol/chloroform extraction after the enzymatic digestion step with micrococcal nuclease (MNase: NEB M0247S). We have added this information in the Materials and Methods section (lines 770-771).

4) For ChIP-qPCR you used mock IgG as a control "Control IgG represents the control for the ChIP specificity" (legend fig1). Why not use it also in ChIP-seq?

Our response: We thank the reviewer for this comment. Both nonspecific immunoglobulin G (IgG) antibodies and input chromatin could be used as controls in ChIP-seq experiments. IgG antibodies may be less desirable because IgG antibodies usually immunoprecipitate much less DNA than specific antibodies do, and thus resulting overamplifying during the library construction step⁽¹⁾. In this study, we attempted to perform ChIP-seq by using IgG antibody but obtained a low amount of ChIP DNA (less than 1 ng). In the meantime, we also found that input DNA was commonly used as controls in ChIP-seq experiments for histone modifications in fungi, oomycetes, plants, and animals systems⁽²⁻⁷⁾. Therefore, we used input DNA instead of IgG ChIP DNA as controls for ChIP-seq experiments in this study.

REFERENCES

(1) Kidder, B. L., Hu, G. & Zhao, K. ChIP-Seq: technical considerations for obtaining high-quality data. *Nat. Immunol.* **12**, 918-22 (2011).

- (2) Wang, L., Chen, H., Li, J., Shu, H., Zhang, X., Wang, Y., Tyler, B. M. & Dong, S. Effector gene silencing mediated by histone methylation underpins host adaptation in an oomycete plant pathogen. *Nucleic Acids Res.* **48**, 1790-1799 (2020).
- (3) Martin, B. J. E., Brind'Amour, J., Kuzmin, A., Jensen, K. N., Liu, Z. C., Lorincz, M. & Howe, L. J. Transcription shapes genome-wide histone acetylation patterns. *Nat. Commun.* **12**, 210 (2021).
- (4) Wang, C., Zhang, L., Ke, L., Ding, W., Jiang, S., Li, D., Narita, Y., Hou, I., Liang, J., Li, S., Xiao, H., Gottwein, E., Kaye, K. M., Teng, M. & Zhao, B. Primary effusion lymphoma enhancer connectome links super-enhancers to dependency factors. *Nat. Commun.* **11**, 6318 (2020).
- (5) Cattaneo, P., Hayes, M. G. B., Baumgarten, N., Hecker, D., Peruzzo, S. et al. DOT1L regulates chamber-specific transcriptional networks during cardiogenesis and mediates postnatal cell cycle withdrawal. *Nat. Commun.* **13**, 7444 (2022).
- (6) Chen, X., MacGregor, D. R., Stefanato, F. L., Zhang, N., Barros-Galvão, T. & Penfield, S. A VEL3 histone deacetylase complex establishes a maternal epigenetic state controlling progeny seed dormancy. *Nat. Commun.* **14**, 2220 (2023).
- (7) Tang, G., Yuan, J., Wang, J., Zhang, Y. Z., Xie, S. S., Wang, H., Tao, Z., Liu, H., Kistler, H. C., Zhao, Y., Duan, C. G., Liu, W., Ma, Z. & Chen, Y. *Fusarium* BP1 is a reader of H3K27 methylation. *Nucleic Acids Res.* **49**, 10448-10464 (2021).

5) How did you normalize MNase-qPCR assays? I'm not convinced (yet) that what you are showing is true. Accurate quantification is a tricky thing.

Our response: Sorry for missing the details. The relative nucleosome occupancy was determined by tiled qPCR analysis as described previously^(1,2). The relative nucleosome occupancy was calculated by normalizing signals from the MNase-treated DNA to the corresponding undigested genomic DNA of each sample. At the end, all values were further normalized to that of *FgACTIN* 100+ loci for each sample. We have now included this information in the Materials and Methods section section (Lines 871-874).

REFERENCES

(1) Kaster, M. & Laubinger, S. Determining nucleosome position at individual loci after biotic stress using MNase-qPCR. *Methods Mol. Biol.* **1398**, 357–372 (2016).

(2) Tang, G., Yuan, J., Wang, J., Zhang, Y. Z., Xie, S. S., Wang, H., Tao, Z., Liu, H., Kistler, H. C., Zhao, Y., Duan, C. G., Liu, W., Ma, Z. & Chen, Y. *Fusarium* BP1 is a reader of H3K27 methylation. *Nucleic Acids Res.* **49**, 10448-10464 (2021).

6) To ensure data FAIRness, please give parameters of softs you used in a supplemental table for example.

Our response: As suggested, we have provided the parameters of softs in Supplementary Table 2 and have added this information in Materials and Methods section (lines 892-894).

- I must say I'm not very much at ease with the ChIP profiles shown in supp fig 1. The *in vitro* H3K27me3 is not quite as expected from what is known, i.e., H3K27me3 marks very strong and localized islands in all 4 chromosomes (see Connolly et al 2013). Connolly's pattern has been extensively proved and reproduced in various Fg labs. Any idea on why we do not observe this here? What could be the consequences on the bivalence you propose? Please develop and discuss.

Our response: We are sorry for the confusion here. In our previous version Supplementary Fig. 1b, the library-size normalized BPM data range for H3K27me3 modification was erroneously set too large due to the presence of outliers, resulting in a difference of our *in vitro* H3K27me3 profile compared to Connolly's study⁽¹⁾. For better visualization of the H3K4me3/H3K27me3 pattern in the IGV browser, we have properly adjusted the data range without considering outliers (outliers accounted for less than 1% of each sample). Data range adjustment did not affect the ChIP-seq analysis in this study. Accordingly, we have updated Supplementary Fig. 1b in the revised manuscript. It showed that *in vitro* H3K27me3 marks in our study also exhibit strong and localized islands across all four chromosomes (Supplementary Fig. 1b in the revised manuscript). To further address the reviewer's concern, we compared the H3K27me3 pattern from this study with a recent study by Guangfei Tang et al⁽²⁾ using

the similar methods for ChIP-seq analysis (see below Fig. 2). The results showed that H3K27me3 pattern in our study is similar to that in Guangfei Tang's paper, further confirming the reliability of our results. We have described this in line 119 of the revised manuscript.

Fig. 2 A comparison of the *in vitro* ChIP-seq signals of H3K27me3 within four chromosomes of *F. graminearum* between our study and Guangfei Tang's study. In the IGV browser, the H3K27me3 signal from this paper is displayed in light blue, while Guangfei Tang's study is represented in deep blue. The Y-axis represents the ChIP-seq signal density, which was calculated using BPM (Bins Per Million mapped reads) with 50-bp resolution.

REFERENCES

- (1) Connolly, L. R., Smith, K. M. & Freitag, M. The *Fusarium graminearum* histone H3 K27 methyltransferase KMT6 regulates development and expression of secondary metabolite gene clusters. *PLoS Genet.* **9**, e1003916 (2013).
- (2) Tang, G., Yuan, J., Wang, J., Zhang, Y. Z., Xie, S. S., Wang, H., Tao, Z., Liu, H., Kistler, H. C., Zhao, Y., Duan, C. G., Liu, W., Ma, Z. & Chen, Y. *Fusarium* BP1 is a reader of H3K27 methylation. *Nucleic Acids Res.* **49**, 10448-10464 (2021).

- How did FgACTIN perform as an internal reference for qPCR gene expression analyses? Best practices are to use more than one gene as internal reference and evaluate their validity, using genorm or equivalent for example (and I believe genorm

is built in QuantStudio suite, so it should be an easy thing to do to test several housekeeping genes and make sure they behave well).

Our response: We thank the reviewer for the comments. Following the reviewer's suggestion, we have evaluated five candidate genes (*FgACTIN*, *FgTubulin*, *FgGADPH*, *FgEF1a*, *FgUBH*) that are most commonly used as reference genes in *Fusarium graminearum*¹⁻⁴. Fig. 1a showed their M values according to geNorm analysis by using NormqPCR packages^(5,6) (see below Fig. 3a). GeNorm analysis indicated an average expression stability M ranging from 0.16 using all five genes to 0.059 for the two most stable candidates, *FgACTIN* and *FgTubulin* (see below Fig. 3b). Furthermore, geNorm also computed variation of the reference genes used for normalization (GeNorm V: $n/(n+1)$). We found that V2/3 showed a V value of 0.046, below the threshold of 0.15 (see below Fig. 3b). Therefore, the most suitable reference genes for evaluating *BCG1* expression are *FgACTIN* and *FgTubulin*. To further address the reviewer's concern, we also determined the expression of *BCG1* during host infection using *FgTubulin* as the reference gene via RT-qPCR assay. As shown in below Fig. 4, we got similar expression pattern of *BCG1*, further confirming the reliability of the RT-qPCR results in this study using *FgACTIN* as an internal reference gene. We have described this in the revised manuscript (lines 821-822).

Fig. 3 Evaluation of overall stability of five candidate reference genes using geNorm analysis. **a** geNorm analysis of five candidate reference genes. Average expression stability (M) of the remaining candidates after stepwise removal of the least stable gene is shown. The least stable gene in each step is indicated below. **b** geNorm pairwise variation (V) analysis to determine the sufficient number of reference genes in the normalization factor.

Fig. 4 Relative expression levels of *BCG1* in wild-type PH-1 and the indicated mutants at 0-120 hours post-inoculation (hpi) were evaluated via RT-qPCR by using *FgTubulin* as an internal reference gene. Data presented are the means \pm standard deviation (SD) (n=3 biological replicates).

REFERENCES

- (1) Jian, Y., Liu, Z., Wang, H., Chen, Y., Yin, Y., Zhao, Y. & Ma, Z. Interplay of two transcription factors for recruitment of the chromatin remodeling complex modulates fungal nitrosative stress response. *Nat. Commun.* **12**, 2576 (2021).
- (2) Jia, L. J., Tang, H. Y., Wang, W. Q., Yuan, T. L., Wei, W. Q., Pang, B., Gong, X. M., Wang, S. F., Li, Y. J., Zhang, D., Liu, W. & Tang, W. H. A linear nonribosomal octapeptide from *Fusarium graminearum* facilitates cell-to-cell invasion of wheat. *Nat. Commun.* **10**, 922 (2019).
- (3) Yu, J., Park, J. Y., Heo, J. I. & Kim, K. H. The ORF2 protein of *Fusarium graminearum* virus 1 suppresses the transcription of FgDICER2 and FgAGO1 to limit host antiviral defences. *Mol. Plant Pathol.* **21**, 230-243 (2020).
- (4) Nasmith, C. G., Walkowiak, S., Wang, L., Leung, W. W., Gong, Y., Johnston, A., Harris, L. J., Guttman, D. S., & Subramaniam, R. Tri6 is a global transcription regulator in the phytopathogen *Fusarium graminearum*. *PLoS Pathog.* **7**, e1002266 (2011).
- (5) McDermott, A. M., Kerin, M. J. & Miller, N. Identification and validation of miRNAs as endogenous controls for RQ-PCR in blood specimens for breast cancer studies. *PLoS ONE* **8**, e83718 (2013).
- (6) Vandesompele, J., De Preter, K., Pattyn, F., Poppe, B., Van Roy, N., De Paepe, A. & Speleman, F. Accurate normalization of real-time quantitative RT-PCR data by geometric averaging of multiple internal control genes. *Genome Biol.* **3**, research0034.1 (2002).

- For dotblot validations, please provide alignments of sequences in supp data showing that the peptides you used match the ones found in Fig.

Our response: We thank the reviewer for the comments. As suggested, we have provided the alignments of histone H3 protein in *F. graminearum* (1-100 residues) with the sequences of H3K4me1/2/3 and H3K27me1/2/3 peptides in supplementary Fig. 19a.

- Is figure 1 H3K4m3 then H3K27me3 or the reverse? Or the intersect? Please clarify.

Our response: We are sorry for the confusion here. K4-K27 represents sequential ChIP conducted by first ChIP using an anti-H3K4me3 antibody followed by second ChIP using an anti-H3K27me3 antibody, while K27-K4 represents sequential ChIP performed using antibodies in the reverse order. We have revised Fig. 1d and the figure legend of Fig. 1 c and d in the revised manuscript (lines 913-916).

- The mutants are still a bit of a sticky point to me. Have you checked “some phenotypes” for all 3 mutants you made or did you perform add-backs? Best practices recommend caution (and especially when nearly lethal deletions are considered, such as such as $\Delta set1$ and $\Delta kmt6$ in Fg).

Our response: We are sorry for confusions here. In this study, we obtained three deletion mutants for each gene with similar phenotypes. All of the knockout mutants were characterized for defects in wheat infection and vegetative growth (See below Fig. 5a, b). Following the reviewer’s suggestion, we selected the mutants with obvious defects in fungal virulence or vegetative growth ($\Delta Fg08481$, $\Delta Fg05155$, $\Delta Set1$, $\Delta BP1$, and $\Delta Kmt6$) for complementation assays. For all of them, the reintroduction of the wild-type allele rescues the defects observed in the corresponding mutants (See below Fig. 5 a, b). We have included the corresponding information of the mutants in supplementary Fig. 18. Please see lines 640-641.

Fig. 5 All of the knockout mutants in this study were characterized for defects in vegetative growth (a) and wheat infection (b). Different letters indicate significant differences (*p*-value < 0.05, one-way ANOVA).

Reviewer #4 (Remarks to the Author):

I am happy with this revision, No further questions.

Our response: We thank the reviewer for the very positive evaluation of our revised manuscript and the constructive comments on the manuscript.